  

# A unique bacterial tactic to circumvent the cell death crosstalk induced by blockade of caspase-8

Hiroshi Ashida[1,2,*] (ID), Chihiro Sasakawa[2,3] (ID) & Toshihiko Suzuki[1,**] (ID)

## Abstract

Upon invasive bacterial infection of colonic epithelium, host cells induce several types of cell death to eliminate pathogens. For instance, necroptosis is a RIPK-dependent lytic cell death that serves as a backup system to fully eliminate intracellular pathogens when apoptosis is inhibited; this phenomenon has been termed "cell death crosstalk". To maintain their replicative niche and multiply within cells, some enteric pathogens prevent epithelial cell death by delivering effectors via the type III secretion system. In this study, we found that *Shigella* hijacks host cell death crosstalk via a dual mechanism: inhibition of apoptosis by the OspC1 effector and inhibition of necroptosis by the OspD3 effector. Upon infection by *Shigella*, host cells recognize blockade of caspase-8 apoptosis signaling by OspC1 effector as a key danger signal and trigger necroptosis as a backup form of host defense. To counteract this backup defense, *Shigella* delivers the OspD3 effector, a protease, to degrade RIPK1 and RIPK3, preventing necroptosis. We believe that blockade of host cell death crosstalk by *Shigella* is a unique intracellular survival tactic for prolonging the bacterium's replicative niche.

**Keywords** apoptosis; effector; necroptosis; *Shigella*

**Subject Categories** Autophagy & Cell Death; Microbiology, Virology & Host Pathogen Interaction

The EMBO Journal (2020) 39: e104469

See also: **SJ Thygesen *et al*** (September 2020)

## Introduction

Epithelial cell death is an intrinsic immune defense against bacterial intrusion. The sacrifice of infected cells plays an important role in clearance of damaged cells, elimination of pathogens, and presentation of bacteria-derived antigens to the adaptive immune system

(Yuan *et al*, 2016; Jorgensen *et al*, 2017). Because the host induces several types of cell death (e.g., apoptosis, necrosis, pyroptosis, and necroptosis) depending on the stage of infection, intensity of infection, host cell type, and physiological state, cell death is considered to be a key aspect of host defense against bacterial intrusion (Lamkanfi & Dixit, 2010; Rudel *et al*, 2010). Apoptosis is a caspase-dependent, non-inflammatory form of cell death triggered by a mitochondria-mediated or receptor-mediated pathway, whereas pyroptosis and necroptosis are lytic forms of cell death that require caspase-1/-4–GSDMD and RIPK3–MLKL signaling, respectively (Kerr *et al*, 1972; Holler *et al*, 2000; Degterev *et al*, 2005; Kayagaki *et al*, 2015; Shi *et al*, 2015). Indeed, crosstalk between different cell death modalities plays an important role in host defense (Place & Kanneganti, 2019). For instance, host cells trigger necroptosis as a backup system to overcome apoptosis inhibition by pathogens (Mocarski *et al*, 2015).

Intriguingly, in intestinal epithelial cells infected with bacterial pathogens such as *Shigella*, enteropathogenic *Escherichia coli* (EPEC), and enterohemorrhagic *E. coli* (EHEC), which colonize within or upon epithelia, cell death is not observed (Jones *et al*, 2008; Hemrajani *et al*, 2010; Blasche *et al*, 2013; Kobayashi *et al*, 2013; Li *et al*, 2013; Pearson *et al*, 2013). This suggests that because cell survival is necessary for these bacterial pathogens to maintain their replicative niches, they deploy multiple countermeasures that prevent activation of host cell death pathways (Stewart & Cookson, 2016). In particular, these pathogens deliver effector proteins via the type III secretion system (T3SS), which enables bacterial invasion, regulation of host cell death, and evasion of the immune system, allowing them to efficiently colonize the intestinal epithelium (Pinaud *et al*, 2018).

*Shigella flexneri*, a causative agent of bacillary dysentery, invades and colonizes host epithelial cells, ultimately leading to severe inflammatory colitis. *Shigella* injects a subset of effectors via T3SS into host cells, allowing the bacterium to invade epithelial cells, escape from the vacuole, and multiply within the cytoplasm (Ashida *et al*, 2015). During *Shigella* invasion and multiplication within epithelial cells, the bacteria release pathogen-associated molecular patterns (PAMPs) and create damage-associated molecular patterns (DAMPs), including genotoxic stress, mitochondrial damage, and oxidative stress (Ashida *et al*, 2015). The host cell recognizes these

---

1 Department of Bacterial Infection and Host Response, Graduate School of Medical and Dental Sciences, Tokyo Medical and Dental University (TMDU), Tokyo, Japan
2 Medical Mycology Research Center, Chiba University, Chiba, Japan
3 Nippon Institute for Biological Science, Tokyo, Japan
 *Corresponding author. Tel: +81 3 3813 6111; E-mail: ashi.bact@tmd.ac.jp
 **Corresponding author. Tel: +81 3 3813 6111; E-mail: suzuki.bact@tmd.ac.jp

PAMPs and DAMPs at the early stage of infection and induces several types of cell death as a defense mechanism aimed at terminating the infection (Carneiro *et al*, 2009; Dupont *et al*, 2009; Bergounioux *et al*, 2012). However, *Shigella* delivers a subset of T3SS effectors and prevents epithelial cell death to maintain its replicative scaffold, allowing the bacteria to multiply and spread to neighboring cells, thereby evading immune surveillance (Pendaries *et al*, 2006; Carneiro *et al*, 2009; Bergounioux *et al*, 2012; Ashida *et al*, 2014; Mou *et al*, 2018). For example, in the early stage of infection, *Shigella* prevents caspase-4-dependent pyroptotic cell death by delivering the T3SS effector OspC3 (Kobayashi *et al*, 2013). However, the effectors that circumvent host cell death in the late stage of infection remain poorly understood. Because there are various types of cell death, different pathways involving in cell death, and crosstalk between the various cell death modalities, it is difficult to identify the molecular mechanisms underlying the effector functions that regulate host cell death.

The molecular interplay of cell death crosstalk is an emerging field of research that is crucial for understanding development, homeostatic maintenance, and immune responses (Fritsch *et al*, 2019; Newton *et al*, 2019). The various cell death processes modulate each other via mutual inhibitory mechanisms, and other pathways stand ready to serve as backup routes in the event of a defect in the first-line cell death response. For instance, necroptosis serves as a backup system when apoptosis is inhibited by pharmacological agents, genetic mutation, or infection (Pasparakis & Vandenabeele, 2015).

In this study, we obtained the first evidence that crosstalk between two cell death pathways, apoptosis and necroptosis, at the late stage of infection plays a major role in host defense. We identified a dual inhibitory mechanism in which apoptosis is prevented by the *Shigella* T3SS effector OspC1 and necroptosis by the OspD3 effector. Infected cells recognize blockade of caspase-8 apoptosis signaling by *Shigella* OspC1 as a DAMP and trigger necroptosis as a backup form of host defense. To counteract this defense mechanism, the *Shigella* T3SS effector OspD3 prevents necroptosis by targeting RIPK1 and RIPK3 for degradation. We believe that blockade of host cell death crosstalk by *Shigella* is a unique intracellular survival tactic that helps bacteria to maintain their replicative scaffold.

# Results

### The *Shigella* OspD3 effector inhibits lytic cell death

To elucidate bacterial strategies for counteracting host epithelial cell death, we infected HT29 human colon epithelial cells with *S. flexneri* WT, S325 (T3SS-deficient mutant; non-invasive), or other mutant strains lacking T3SS-secreted effectors. Lactate dehydrogenase (LDH) cytotoxicity assay, an established indicator of lytic cell death, revealed that cells infected with a *Shigella* mutant lacking *ospD3* (Δ*ospD3*) underwent cell death at a higher rate than WT *Shigella* (Fig 1A). *Shigella* OspD has three homologs, OspD1, OspD2, and OspD3 (Tobe *et al*, 2006), but neither Δ*ospD1* nor Δ*ospD2* enhanced cell death, although the Δ*ospD123* (*ospD1, ospD2,* and *ospD3* genes triple mutant) did (Fig 1A).

Cell death induced by bacterial infection can be morphologically classified into two types, non-lytic (e.g., apoptosis) and lytic (e.g.,

necrosis, pyroptosis, and necroptosis) (Ashida *et al*, 2011). Hence, we characterized the morphology and physiology of cell death induced by Δ*ospD3*. TUNEL assays revealed no significant difference between WT and Δ*ospD3* infected cells, indicating that Δ*ospD3* infection did not cause apoptosis (Fig 1B, top). By contrast, propidium iodide (PI) staining, which detects loss of plasma membrane integrity, was significantly higher in Δ*ospD3*-infected cells than in WT-infected cells (Fig 1B, bottom). Giemsa staining confirmed that cells infected with Δ*ospD3* underwent higher levels of membrane rupture (Fig 1C). Together, these data indicate that OspD3 has an activity that prevents lytic cell death.

### *Shigella* Δ*ospD3* induces caspase-independent cell death

Lytic cell death is classified into two forms, caspase-dependent (e.g., pyroptosis) and caspase-independent (e.g., necroptosis) (Jorgensen *et al*, 2017). Pyroptosis is accompanied by activation of caspase-1 or caspase-4, resulting in production of IL-1β and IL-18 (Cookson & Brennan, 2001; Broz & Dixit, 2016). To determine whether lytic cell death induced by Δ*ospD3* is caspase-dependent, we performed cytotoxicity assays in the presence of the pan-caspase inhibitor z-VAD. This treatment failed to block Δ*ospD3*-induced cell death, but did block Δ*ospC3*-induced pyroptosis (Kobayashi *et al*, 2013) (Fig EV1A). Consistent with this, Δ*ospC3* infection caused cleavage of GSDMD and maturation of IL-18, which are hallmarks of pyroptosis, whereas Δ*ospD3* infection did not (Fig EV1B). Accordingly, the levels of LDH release in Δ*ospD3*-infected cells began to increase after the 8-h time point (late stage of infection), whereas Δ*ospC3*-infected cells began to increase after the 2-h time point (early stage of infection) (Fig EV1C). Indeed, *Shigella* Δ*ospD3* had no effect on caspase-1, -3/7, or -8 activities in HT29 cells, suggesting that Δ*ospD3* did not induce caspase-dependent pyroptosis (Fig EV1D). Together, these data suggest that Δ*ospD3*-mediated cell death is different from pyroptosis.

### The *Shigella* effector OspD3 inhibits necroptosis

Because OspD3 did not target caspase-dependent lytic cell death (pyroptosis) (Fig EV1), we next focused on caspase-independent lytic cell death, i.e., necroptosis. Necroptosis is a caspase-independent form of programmed necrosis that requires kinase activity of RIPK1 and RIPK3, followed by phosphorylation of MLKL (Pasparakis & Vandenabeele, 2015). Phosphorylated MLKL binds to the cell membrane and forms a necroptotic pore, inducing cell death (Pasparakis & Vandenabeele, 2015). To determine whether OspD3 is involved in inhibition of necroptosis, we monitored the level of phosphorylated MLKL during *Shigella* infection. When HT29 cells were infected with *Shigella* WT, S325, or *ospD* deletion mutants, the Δ*ospD3* and Δ*ospD123* mutants triggered the phosphorylation of MLKL, whereas the other strains did not (Fig 2A).

To ensure that Δ*ospD3* induces necroptosis, we treated cells with kinase inhibitor or siRNA targeting RIPK1 or RIPK3. Treatment with an RIPK1/RIPK3 inhibitor, but not z-VAD, prevented phosphorylation of MLKL and cytotoxicity in Δ*ospD3*-infected cells (Fig 2B and C). Similarly, siRNA-mediated knockdown of RIPK1 or RIPK3 in HT29 cells infected with Δ*ospD3* decreased levels of phosphorylated MLKL and cytotoxicity to the levels observed in WT infection (Fig 2D and E). These data imply that OspD3 targets and inhibits necroptosis during *Shigella* infection.

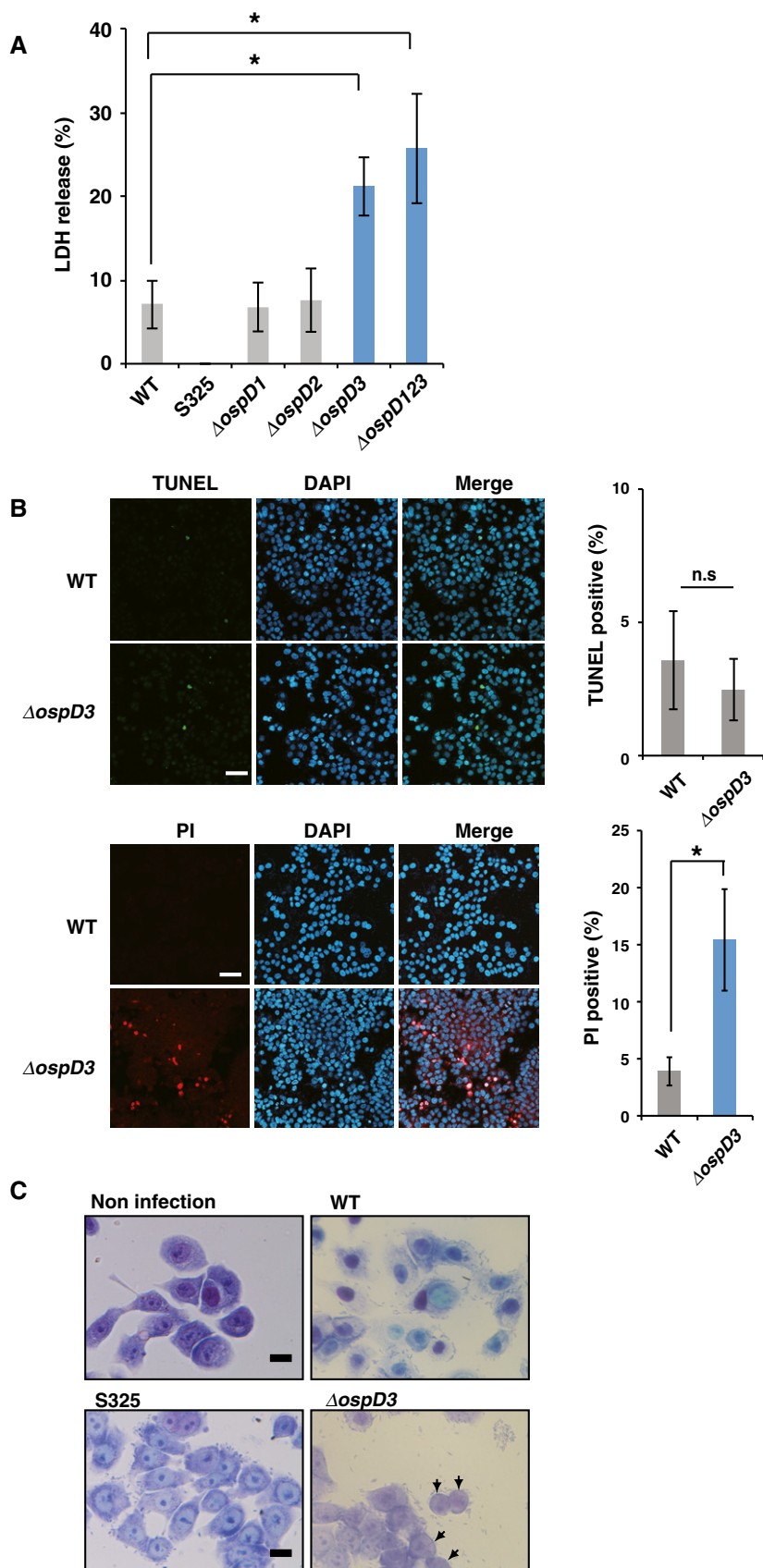

**Figure 1.**

**Figure 1.  *Shigella* OspD3 inhibits lytic cell death.**

A    HT29 cells were infected with *Shigella* WT, S325, or *ospD* deletion mutants and incubated for 8 h. Aliquots of cellular supernatants were subjected to cytotoxicity assays. *P < 0.05 (one-way ANOVA).

B    HT29 cells were infected with *Shigella* WT or Δ*ospD3* and incubated for 8 h. Infected cells were fixed and subjected to TUNEL and PI staining. Percentages of positive cells (TUNEL, green; PI, red) are shown in graph at right. The nuclei were stained with DAPI (blue). Scale bar: 100 μm. n.s., not significant; *P < 0.05 (unpaired two-tailed Student's *t*-test).

C    HT29 cells were infected with *Shigella* WT, S325, or Δ*ospD3* and incubated for 8 h. Infected cells were subjected to Giemsa staining. Arrows indicate cells in which the cytoplasm disappeared. Scale bar: 20 μm.

Data information: Graphs in (A and B) show mean ± SD, and data are pooled from three independent experiments performed in triplicates. Images in (B and C) are representative of three independent experiments.

To ensure the role of OspD3 in infected cells, we measured the levels of cytotoxicity and MLKL phosphorylation in HT29, HeLa (cervix epithelial cells), HaCaT (skin keratinocytes), and HCT116 (colon epithelial cells) cells infected with *Shigella* WT, S325, or Δ*ospD3*. Upon infection with Δ*ospD3*, HT29, but not the other cell lines, exhibited higher cytotoxicity and MLKL phosphorylation than cells infected with WT (Fig EV2A and B). Of importance, RIPK3, which is essential for inducing necroptosis, is expressed only in HT29 but not in HeLa, HaCaT, HCT116, or 293T cells (Fig EV2C). Indeed, HeLa cells stably expressing RIPK3 (HeLa/RIPK3) infected with Δ*ospD3* exhibited higher levels of MLKL phosphorylation and cytotoxicity than in the cells infected with WT, whereas no difference was observed in HeLa cells stably expressing GFP (HeLa/GFP), used as a negative control (Fig EV2D and E). Taken together, we conclude that OspD3 specifically targets and inhibits RIPK-dependent necroptosis during *Shigella* infection.

**OspD3 inhibits necroptosis by degrading RIPK1 and RIPK3**

Because *Shigella* OspD3 inhibits necroptotic death of infected cells, we sought to determine the host target factors involved in necroptosis. To this end, we first investigated necroptosis signaling factors in HT29 cells infected with a series of *Shigella ospD* deletion mutants. Levels of RIPK1 and RIPK3, but not those of the family member RIPK2, were dramatically reduced in cells infected with WT, Δ*ospD1*, or Δ*ospD2*, but not in cells infected with Δ*ospD3* or Δ*ospD123*, indicating an OspD3-dependent decrease of these proteins (Fig 3A). OspD3 homolog effectors are also conserved in EPEC and EHEC (Tobe *et al*, 2006). Consistent with *Shigella* data, the levels of RIPK1 and RIPK3 homologs were dramatically reduced in cells infected with WT EPEC or WT EHEC, but not cells infected with the *ospD3* homolog deletion mutant EPEC Δ*espL* or EHEC Δ*espL2* (Fig EV3A). Intriguingly, unlike cells infected with *Shigella* Δ*ospD3*, cells infected with EPEC Δ*espL* or EHEC Δ*espL2* did not exhibit higher levels of MLKL phosphorylation and cytotoxicity than cells infected with WT (Fig EV3A and B). These results suggest that EPEC and EHEC use redundant pathways to inhibit MLKL phosphorylation and necroptosis and that preventing the degradation of RIPK1 and RIPK3 by deletion of *espL* is insufficient to restore MLKL phosphorylation.

To confirm the effect of OspD3 on RIPK1 degradation, we transfected 293T cells with empty vector or plasmids expressing *ospD1*, *ospD2*, or *ospD3* and measured the levels of RIPK proteins by immunoblotting. As expected, the level of RIPK1, but not RIPK2, was significantly reduced in cells expressing OspD3, but no difference in cells expressing OspD1 or OspD2 (Fig 3B).

Because OspD3 homologs specifically decreased the levels of RIPK1 and RIPK3, we hypothesized that these proteins have protease activity. EspL, the OspD3 homolog of EPEC, has cysteine protease activity dependent on a cysteine protease motif (Cys-His-Asp: C-H-D) (Pearson *et al*, 2017). Importantly, sequence alignment of the OspD3 homologs *Shigella* OspD3, EPEC EspL, and EHEC EspL2 revealed that the C-H-D motif is highly conserved (Fig 3C). To evaluate the functional importance of the C-H-D motif in OspD3, we constructed mutants in which these three residues were replaced by alanine and examined their effects on RIPK1 levels. As shown in Fig 3D, 293T cells expressing OspD3 cleaved RIPK1, whereas cells expressing motif-substituted mutants did not, confirming the importance of the C-H-D motif for protease activity. The C-H-D motifs in OspD3 homologs of EPEC and EHEC were also essential for RIPK1 cleavage (Fig EV3C).

To further confirm the involvement of OspD3 protease activity in *Shigella* infection, we measured the levels of cytotoxicity and MLKL phosphorylation in HT29 cells infected with *Shigella* WT, S325, Δ*ospD3*, Δ*ospD3/D3* (Δ*ospD3* complemented with wild-type *ospD3*), or Δ*ospD3/D3CS* (Δ*ospD3* complemented with a protease activity-deficient mutant, in which the cysteine residue at position 64 was replaced by serine). The levels of RIPK1 and RIPK3 were reduced in cells infected with *Shigella* WT or Δ*ospD3/D3*, but not Δ*ospD3* or Δ*ospD3/D3CS* (Fig 3E). Consistent with OspD3-dependent RIPK cleavage, the levels of MLKL phosphorylation and cytotoxicity were reduced in cells infected with *Shigella* WT or Δ*ospD3/D3*, but not Δ*ospD3* or Δ*ospD3/D3CS* (Fig 3E and F). Taken together, these data indicate that OspD3 targets RIPK1 and RIPK3 for degradation via its protease activity, thereby preventing necroptosis.

**OspD3 targets and cleaves the RHIM domain of RIPK1 and RIPK3**

To obtain more insight into the mechanism by which OspD3 degrades RIPK1 and RIPK3, we constructed a series of RIPK1 truncations and sought to identify the sites of cleavage by OspD3. RIPK1 has three characteristic domains: the N-terminal kinase domain, intermediate domain containing the RIP homotypic interaction motif (RHIM), and C-terminal death domain (He & Wang, 2018). Using the truncations, we investigated OspD3 protease activity-mediated RIPK1 cleavage in 293T cells. OspD3 cleaved RIPK1 truncations containing the RHIM motif (amino acids 532–547), but failed to cleave variants that lacked this sequence (Fig EV4A). The RHIMs in RIPK1 and RIPK3 enable the two proteins to interact and are required for induction of necroptosis (He & Wang, 2018). We therefore constructed a series of amino acid-substituted mutations in the RHIM motif and found that OspD3 targets IQIG (amino acids 539–

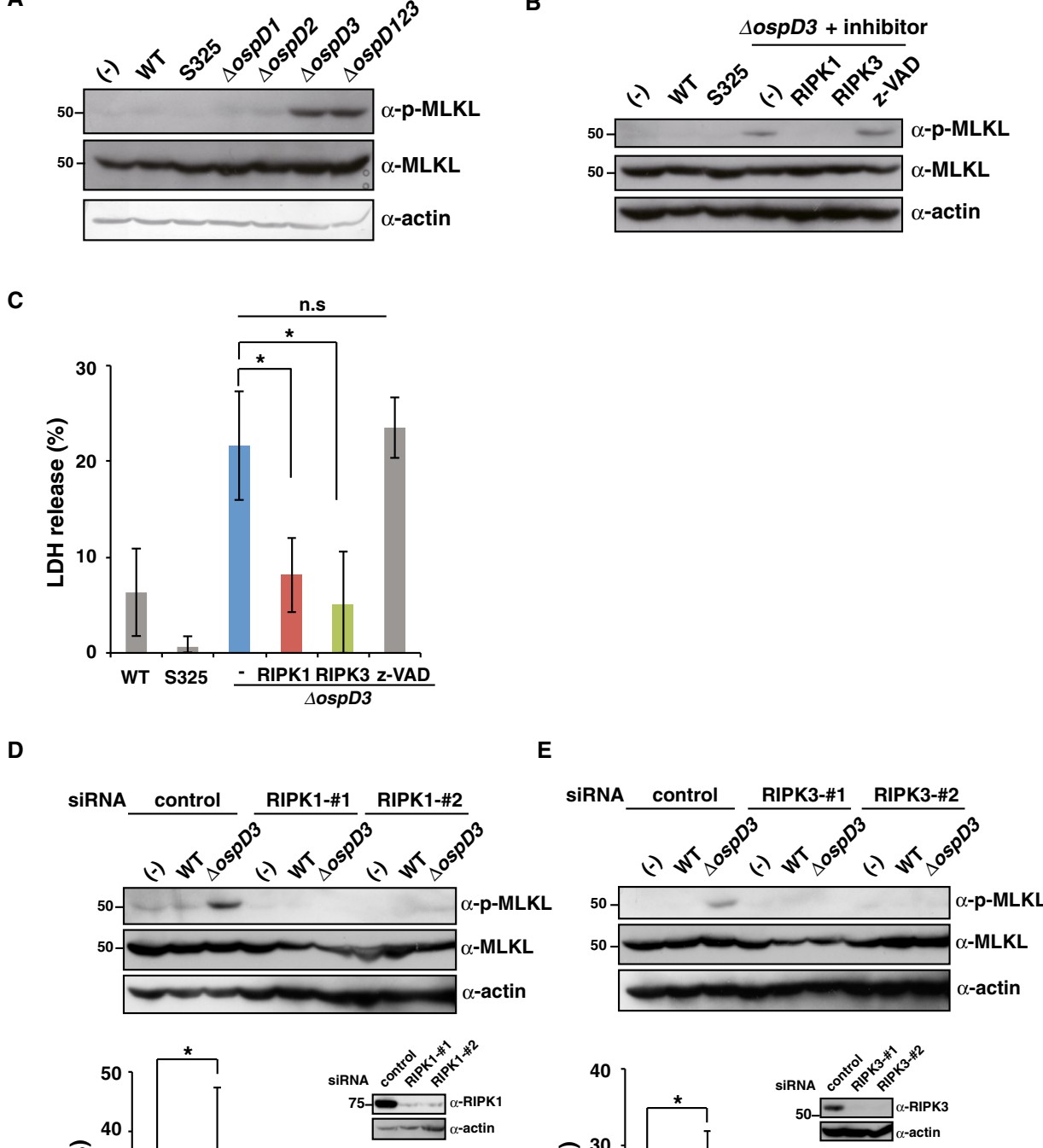

Figure 2.

◀

**Figure 2.** *Shigella* OspD3 inhibits necroptosis.

A   HT29 cells were infected with *Shigella* WT, S325, or *ospD* deletion mutants and incubated for 8 h. Cell lysates were subjected to immunoblotting.

B, C   HT29 cells were infected with the indicated *Shigella* strains in the presence or absence of RIPK1 inhibitor, RIPK3 inhibitor, or caspase inhibitor (z-VAD) and then incubated for 8 h. Cell lysates and aliquots of cellular supernatants were subjected to immunoblotting (B) and cytotoxicity assay (C), respectively. *P < 0.05, n.s., not significant (one-way ANOVA).

D, E   HT29 cells treated with the indicated siRNAs were infected with *Shigella* WT or *ΔospD3*. Cell lysates and aliquots of cellular supernatants were subjected to immunoblotting (top) and cytotoxicity assays (bottom), respectively. The knockdown efficiency of the indicated siRNAs was assessed by immunoblotting (inset). n.s., not significant; *P < 0.05 (unpaired two-tailed Student's *t*-test).

Data information: Graphs in (C–E) show mean ± SD, and data are pooled from three independent experiments performed in triplicates. Images in (A), (B), (D), and (E) are representative of three independent experiments. Molecular weights in immunoblots are in kDa.

Source data are available online for this figure.

542) for cleavage (Fig EV4B). Consistent with this, OspD3 cleaved RIPK1 and RIPK3, but not the RHIM domain mutants RIPK1 (RIPK1-4A) and RIPK3 (RIPK3-4A) (Fig EV4C). Indeed, when HeLa cells stably expressing RIPK3 (HeLa/RIPK3) or RIPK3-4A (HeLa/RIPK3-4A) were infected with *Shigella* WT or *ΔospD3*, the levels of RIPK3, but not RIPK3-4A, were significantly lower in cells infected with *Shigella* WT, but not in those infected with *ΔospD3* (Fig EV4D). Based on these data, we conclude that OspD3 targets the RHIM domain of RIPK1 and RIPK3 for degradation.

## OspC1 inhibits caspase-8 activation and apoptotic cell death

According to the results described above, *Shigella* infection of human epithelial cells triggers necroptosis, and *Shigella* deploys a countermeasure to prevent necroptotic cell death by delivering the protease effector OspD3. Previous studies indicated that when *Shigella* invades and multiplies within epithelial cells, it triggers genotoxic stress, mitochondrial damage, oxidative stress, and recognition of PAMPs and DAMPs, all of which can induce cell death (Carneiro *et al*, 2009; Dupont *et al*, 2009; Bergounioux *et al*, 2012). Therefore, we next sought to identify the events that induce necroptosis during *Shigella* infection. Necroptosis is induced by stimulation via several receptors (e.g., TNF receptor), Toll-like receptors (e.g., TLR3), and intracellular sensors (e.g., DAI). At the same time, caspase-8 plays an important role in induction of necroptosis (Holler *et al*, 2000; Degterev *et al*, 2005). Normally, caspase-8 triggers apoptosis by activating caspase-3 and caspase-7, but it also cleaves RIPK1 and RIPK3, thereby inhibiting necroptosis. If caspase-8 is blocked by pharmacological or genetic mutations, RIPK1 and RIPK3 become phosphorylated, resulting in induction of necroptosis. Thus, caspase-8 determines the type of death that the cell undergoes, i.e., apoptosis or necroptosis.

Because apoptosis is a crucial host defense system, inhibition of apoptosis is a strategy that several bacterial pathogens use to their own advantage (Rudel *et al*, 2010; Robinson & Aw, 2016). Given that several reports have shown that *Shigella* prevents apoptosis in order to maintain the replicative niche (Pendaries *et al*, 2006; Faherty & Maurelli, 2009; Sukumaran *et al*, 2010; Günther *et al*, 2020), we hypothesized that inhibition of caspase-8 activity by *Shigella* might trigger induction of necroptosis. To verify our hypothesis, we measured caspase-8 activity in HT29 cells infected with various effector deletion mutants. As shown in Appendix Fig S1A, infection with *ΔospC123* (*ospC1*, *ospC2*, and *ospC3* triple mutant) significantly increased caspase-8 activity relative to infection with WT. Of importance, among OspC family effectors, OspC3

inhibits caspase-4-dependent pyroptotic cell death, whereas the functions of OspC1 and OspC2 remain unclear (Kobayashi *et al*, 2013). The *ΔospC123* mutant greatly increased cleavage of caspase-8, caspase-3, caspase-7, and PARP1 (a caspase substrate involved in apoptosis signaling) relative to WT, as demonstrated by immunoblotting (Appendix Fig S1B). These results suggest that the OspC123 effectors (OspC1, C2, and C3), independently or synergistically, prevent caspase-8 activation during *Shigella* infection.

To determine which of the OspC effectors prevents caspase-8 activation, we measured caspase-8 activity in HT29 cells infected with each *ospC* gene deletion mutant. As shown in Fig 4A and B, the *ΔospC1* mutant, but neither *ΔospC2* nor *ΔospC3*, greatly increased caspase-8 activation relative to WT. On the other hand, *ΔospC3*, but neither *ΔospC1* nor *ΔospC2*, increased cytotoxicity, cleavage of GSDMD, and maturation of IL-18, which are hallmarks of pyroptosis (Broz *et al*, 2020; Fig 4B and C). Consistently, immunofluorescence revealed that cleaved caspase-8 increased significantly upon *ΔospC1* infection in comparison with WT infection (Fig 4D). Because caspase-8 acts upstream of caspase-3 or caspase-7 in induction of apoptosis, we next measured caspase-3/7 activity and apoptosis in cells infected with *Shigella* WT or *ΔospC1*. Caspase-3/7 activity was significantly higher in cells infected with *ΔospC1* than in those infected with WT, leading to apoptotic cell death (Fig 4E and F). Although the stimuli that directly drive caspase-8 activation remain unclear, these findings corroborate the idea that OspC1 effector can prevent caspase-8 activity during *Shigella* infection, ultimately inhibiting apoptosis.

### *Shigella* T3SS effector-mediated inhibition of apoptosis triggers necroptosis

Because the OspC1 effector prevents caspase-8 activation, a key checkpoint in the cell fate decision between apoptosis and necroptosis, we further pursued the correlation between OspC1-mediated caspase-8 inhibition and induction of necroptosis. To this end, we measured the levels of MLKL phosphorylation and cytotoxicity in HT29 cells infected with *Shigella* WT, S325, *ΔospC1*, *ΔospD3*, *ΔospC1ΔospD3*, or *ΔospC1ΔospD3/ospC1* (*ΔospC1ΔospD3* complemented with *ospC1*). Consistent with the results shown in Figs 1A and 2A, the levels of MLKL phosphorylation and cytotoxicity were higher in cells infected with *ΔospD3* than in cells infected with WT or *ΔospC1*, indicating that *ΔospD3* infection triggers necroptosis (Fig 5A and B). In contrast to infection with *ΔospD3*, infection with *ΔospC1ΔospD3* did not lead to phosphorylation of MLKL or induction of rapid cell

**A**

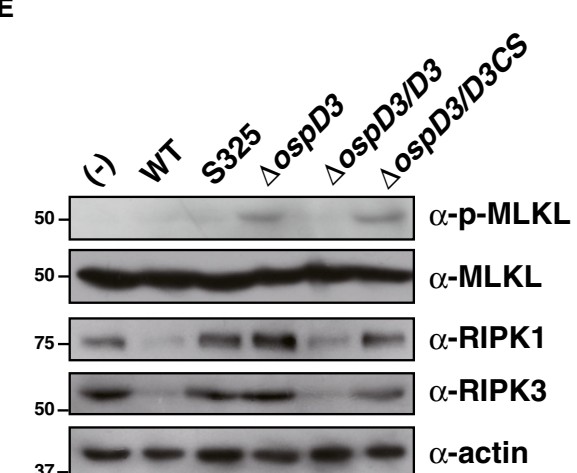

75 — α-RIPK1

75 — α-p-RIPK1

50 — α-RIPK2

50 — α-RIPK3

37 — α-actin

(−)   WT   S325   ΔospD1   ΔospD2   ΔospD3   ΔospD123

**B**

vector   OspD1   OspD2   OspD3

75 — α-RIPK1

50 — α-RIPK2

α-Myc

α-actin

**C**

```
OspD3  61 LIVCRHLASYWIAQFNKSSGHVDYHHFAFPDEIKNYVSVS 100
EPEC   44 IIWCRHIASYWSEFFCSNSGKIDYETFSSPQLLSKAIVIQ 83
EHEC   44 IIWCRHIASYWSEFFCSNSGKIDYETFSSPQLLSKAIVIQ 83
          * *** **** *   *  ** **  *  *

OspD3 101 EEEKAINVPAIIYFVENGSWGDIIFYIFNEMIFHSEKSRA 140
EPEC   84 ENKGTNNIKGDVYFVENESWGSVIYNLFLQLEKENKSHTS 123
EHEC   84 ENKGTNNIKGDVYFVENESWGSVIYNLFLQLEKENKSHTS 123
          *      *      *****  ***    *    *

OspD3 141 LEISTSNHNMALGLKIKETKNGGDFVIQLYDPNHTATHLR 180
EPEC  124 LEVHSPGHAMALGIKIKNDK-ENKFVINFYDPNQTATHKR 162
EHEC  124 LEVHSPGHAMALGIKIKNDK-ENKFVINFYDPNQTATHKR 162
          **     * **** ***  *   ***  ****  **** *
```

**D**

vector   OspD3   OspD3-CA   OspD3-HA   OspD3-DA

75 — α-RIPK1

50 — α-RIPK2

α-actin

**E**

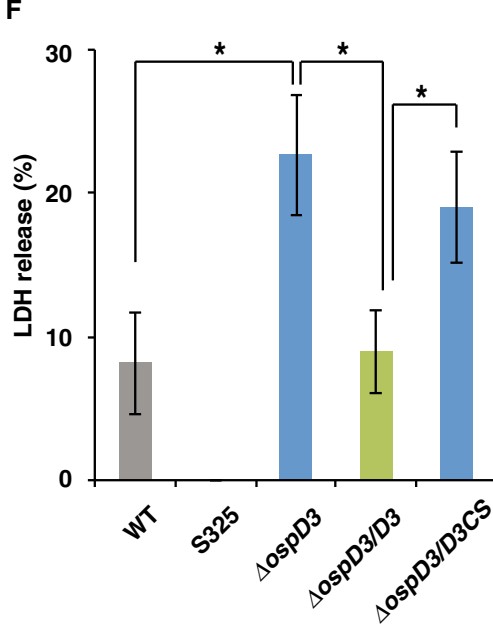

50 — α-p-MLKL

50 — α-MLKL

75 — α-RIPK1

50 — α-RIPK3

37 — α-actin

(−)   WT   S325   ΔospD3   ΔospD3/D3   ΔospD3/D3CS

**F**

LDH release (%)

WT   S325   ΔospD3   ΔospD3/D3   ΔospD3/D3CS

**Figure 3.**

**Figure 3.  OspD3 inhibits necroptosis by degradation of RIPK1 and RIPK3.**

A HT29 cells were infected with the indicated *Shigella* strains and incubated for 8 h, and then, cell lysates were subjected to immunoblotting.

B 293T cells were transfected with Myc-tagged *ospD* expression plasmids. After 24 h, cells were harvested and subjected to immunoblotting. Arrows indicate cleaved RIPK1.

C Multiple sequence alignment of the OspD family: *Shigella* OspD3, EPEC EspL, and EHEC EspL2. Conserved amino acids are indicated by asterisks. Typical protease catalytic sites are colored in red.

D 293T cells were transfected with a series of plasmids expressing *ospD3* point mutants. After 24 h, cells were harvested and subjected to immunoblotting. Arrows indicate cleaved RIPK1.

E, F HT29 cells were infected with *Shigella* WT, S325, *ΔospD3*, *ΔospD3/D3* (*ΔospD3* complemented with wild-type *ospD3*), or *ΔospD3/D3CS* (*ΔospD3* complemented with a protease activity-deficient mutant, in which the cysteine residue at position 64 was replaced by serine) strains and incubated for 8 h. Cell lysates and aliquots of cellular supernatants were subjected to immunoblotting (E) and cytotoxicity assays (F), respectively. *$P < 0.05$ (one-way ANOVA).

Data information: Graph in (F) shows mean $\pm$ SD, and data are pooled from three independent experiments performed in triplicates. Images in (A), (B), (D), and (E) are representative of three independent experiments. Molecular weights in immunoblots are in kDa.

Source data are available online for this figure.

death, whereas caspase-8 activity was elevated in cells infected with *ΔospC1* or *ΔospC1ΔospD3* (Fig 5A and B). Intriguingly, the reduction in the levels of MLKL phosphorylation and cytotoxicity in cells infected with *ΔospC1ΔospD3* was rescued by *ospC1* gene complementation (*ΔospC1ΔospD3/ospC1*), indicating that OspC1-mediated caspase-8 inhibition induced necroptosis (Fig 5A and B). Cells infected with the *ΔospC2ΔospD3* or *ΔospC3ΔospD3* double mutant still triggered phosphorylation of MLKL, like cells infected with *ΔospD3*, suggesting an absence of crosstalk between OspC3-mediated pyroptosis inhibition and necroptosis (Fig EV5A). Furthermore, a time course of caspase-8 activation in *ΔospC1*-infected cells revealed that caspase-8 stimulus began to accumulate after 5 h post-infection (Figs 5C and EV5B). Accordingly, the levels of MLKL phosphorylation in *ΔospD3*-infected cells began to increase after the 6-h time point (Figs 5C and EV5B). These data strongly indicate that OspC1-mediated caspase-8 inhibition becomes a cue to trigger necroptosis, which was eventually counteracted by OspD3. By contrast, the time courses of cytotoxicity, IL-18 maturation, and GSDMD cleavage revealed that OspC3 prevented pyroptosis at 2 h post-infection (Figs EV1C and EV5C). Taken together, these data reveal the order of sequential cell death inhibition by *Shigella*; OspC3 acts first to prevent pyroptosis, followed by OspC1-mediated inhibition of apoptosis and OspD3-mediated inhibition of necroptosis.

To further support our hypothesis, we treated cells infected with various *Shigella* mutants with caspase-8 inhibitor to prevent *ΔospC1*-dependent caspase-8 activation. Although the increase in MLKL phosphorylation and cytotoxicity observed in cells infected with *ΔospD3* was abolished in cells infected with *ΔospC1ΔospD3*, caspase-8 inhibitor rescued phosphorylation of MLKL and cytotoxicity in cells infected with *ΔospC1ΔospD3* (Fig 5D and E). Similarly, siRNA-mediated knockdown of caspase-8 in HT29 cells infected with *ΔospC1ΔospD3* restored the levels of phosphorylation of MLKL and cytotoxicity to the levels observed in *ΔospD3* infection (Figs 5F and EV5D). These findings highlight the importance of OspD3-mediated inhibition of crosstalk between apoptosis and necroptosis in prolonging bacterial colonization. Based on the results described above, we assume that necroptosis is induced during *Shigella ΔospD3* infection via inhibition of caspase-8 by OspC1, which is the consequence of a *Shigella* survival strategy that counteracts apoptosis to promote intracellular colonization.

## Discussion

Upon bacterial infection of intestinal epithelial cells, induction of apoptosis is followed by extrusion of the infected cells from the epithelial monolayer, whereas necroptosis leads to cell lysis and release of intracellular contents and inflammatory cytokines. In this study, we identified a unique bacterial stratagem that circumvents host crosstalk between apoptosis and necroptosis during the late stage of *Shigella* infection. We showed that host cells recognize the blockade of caspase-8 apoptosis signaling by *Shigella* OspC1 effector as a DAMP and subsequently trigger necroptosis as a backup form of host defense. To counteract this response to bacterial infection, *Shigella* delivers a protease, the OspD3 effector, that targets RIPK1 and RIPK3 for degradation and prevents necroptosis (Fig 6). Thereby, *Shigella* prevents both apoptosis and necroptosis in order to preserve its replicative niche and promote bacterial multiplication therein. This is the first report showing how and why a bacterial pathogen counteracts host cell death crosstalk to maintain its replicative scaffold in the late stage of epithelial infection.

OspD3 prevents secretion of IL-8 from *Shigella*-infected epithelial cells (Faherty *et al*, 2016). Because RIPK1 is an essential signaling factor in NF-κB-mediated inflammatory signaling, this phenotype is consistent with our data. The phenotype might be the result of OspD3-mediated degradation of RIPK1/RIPK3, leading to inhibition of NF-κB signaling.

In the absence of caspase-8 activity, necroptosis can be triggered by stimulation of death receptor family proteins, such as TNFR or FasL (Pasparakis & Vandenabeele, 2015). Caspase-8 is a key inhibitor of necroptosis that can cleave and inactivate RIPK1 and RIPK3. When caspase-8 is active, it forms a complex with TRADD and FADD, triggering apoptosis. On the other hand, when caspase-8 is inhibited, RIPK1 and RIPK3 are phosphorylated, triggering necroptosis (Holler *et al*, 2000; Degterev *et al*, 2005). Because apoptosis is a major cellular defense response, many bacterial pathogens, including *Shigella*, deploy multiple countermeasures that prevent apoptosis signaling, thereby keeping the host cells alive and promoting full replication within the cells (Ashida *et al*, 2014; Robinson & Aw, 2016).

In this study, we demonstrated that *Shigella* OspC1 blocks caspase-8 activation to prevent apoptosis, whereas host cells recognize OspC1-mediated caspase-8 inhibition and switch to necroptosis from apoptosis. Therefore, blockade of caspase-8-dependent

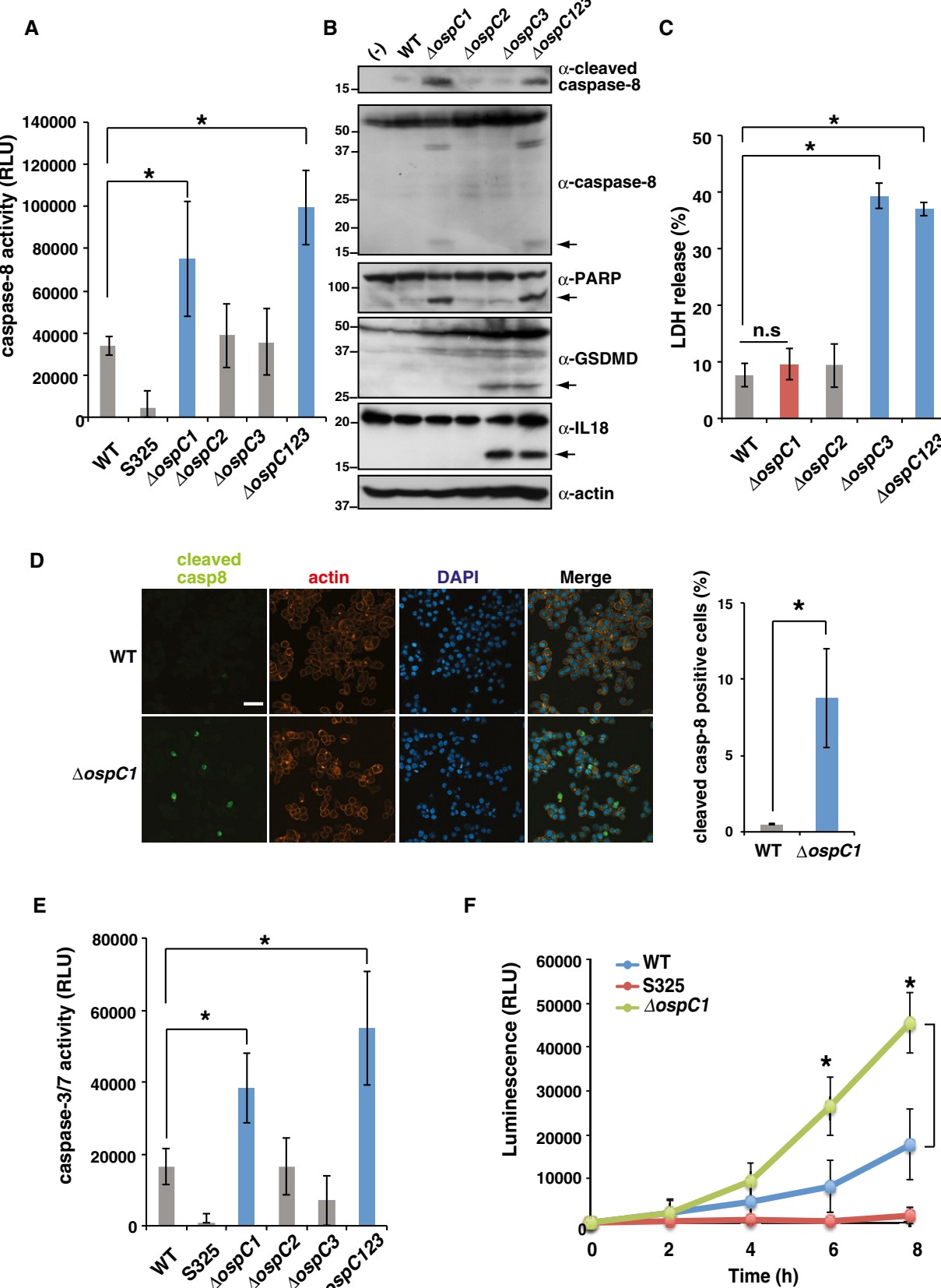

**Figure 4.**

**Figure 4.  *Shigella* OspC1 effector prevents caspase-8 activation.**

A   HT29 cells were infected with the indicated *Shigella* strains and incubated for 8 h. Cells were harvested and subjected to measurement of caspase-8 activation. Caspase-8 activity is reported as relative light units (RLU) of infected samples, minus the value in uninfected samples. *$P < 0.05$ (one-way ANOVA).

B   HT29 cells were infected with the indicated *Shigella* strains and incubated for 8 h. Cell lysates were subjected to immunoblotting. Arrows indicate cleaved forms of caspase-8, PARP, GSDMD, and IL-18, respectively.

C   HT29 cells were infected with the indicated *Shigella* strains and incubated for 8 h. Aliquots of cellular supernatants were subjected to cytotoxicity assay. *$P < 0.05$; n.s., not significant (one-way ANOVA).

D   HT29 cells were infected with *Shigella* WT or *ΔospC1* mutant and incubated for 8 h. Infected cells were then fixed and stained with cleaved caspase-8 (green), rhodamine–phalloidin (red), and DAPI (blue). Percentages of positive cells are shown in the graph at right (*$P < 0.05$; unpaired two-tailed Student's *t*-test). Scale bar: 100 μm.

E   HT29 cells were infected with the indicated *Shigella* strains and incubated for 8 h. Cells were harvested and subjected to measurement of caspase-3/7 activation. Caspase-3/7 activity is reported as relative light units (RLU) of infected samples minus uninfected samples. *$P < 0.05$ (one-way ANOVA).

F   HT29 cells were infected with *Shigella* WT, S325, or *ΔospC1* strains and incubated for up to 8 h. After incubation for the indicated times, cells were subjected to real-time Annexin V apoptosis assay. Annexin V binding is reported as relative light units (RLU) in infected samples minus the value in uninfected samples. *$P < 0.05$ (one-way ANOVA).

Data information: Graphs in (A) and (C)–(F) show mean ± SD, and data are pooled from three independent experiments performed in triplicates. Images in (B) and (D) are representative of three independent experiments. Molecular weights in immunoblots are in kDa.
Source data are available online for this figure.

apoptosis by OspC1 serves as a cue to induce a backup form of host defense (necroptosis), which is eventually counteracted by the protease activity of the OspD3 effector. It remains unclear whether other bacterial pathogens counteract crosstalk between apoptosis and necroptosis. EPEC and EHEC deliver the NleB1/NleB and NleF effectors to antagonize caspase-8 activation and also deliver EspL to prevent necroptosis signaling (Li *et al*, 2013; Pearson *et al*, 2013; Pollock *et al*, 2017). *Shigella* OspC1 is similar to neither EPEC NleB1 nor NleF, whereas *Shigella* OspD3 and EPEC EspL are homologous. Hence, identification of the differences in cell death inhibition strategies between *Shigella* and EPEC is an important priority of future work.

It is tempting to speculate that the cell death crosstalk between apoptosis and necroptosis may have evolved as a host defense against infection by bacteria that possess inhibitors of caspase-8, whereas *Shigella* has evolved a further survival strategy that circumvents cell death crosstalk by counteracting necroptosis. Thus, in the arms race between bacterial pathogens and host defense systems, both pathogens and host deploy offense and defense strategies for their own benefit.

Host cells are able to mobilize countermeasures, such as crosstalk between cell death pathways, by sensing bacterial disturbance of innate immune signaling pathways. For example, host cells detect inhibition of TAK1 and IKK mediated by the *Yersinia* effector YopJ and then trigger caspase-8-dependent GSDMD cleavage, leading to pyroptotic cell death and inflammation (Orning *et al*, 2018; Sarhan *et al*, 2018). In addition, host cells specifically recognize inactivation of RhoA by *Yersinia* effectors YopE and YopT and then activate pyrin inflammasome and pyroptosis to restrict bacterial infection. However, *Yersinia* delivers another T3SS effector, YopM, that specifically suppresses the pyrin inflammasome triggered by YopE- and YopT-mediated inactivation of RhoA (Chung *et al*, 2016; Ratner *et al*, 2016).

Studies indicate that intestinal epithelial cells infected by enteric bacterial pathogens, including *Shigella*, EPEC, EHEC, and *Salmonella*, do not undergo cell death at an early stage of infection. In fact, some enteric bacterial pathogens, which prefer epithelial cells as an infectious scaffold, seem to have evolved sophisticated infectious strategies that counteract host cell death in order to maintain their infectious scaffold until the bacteria have fully multiplied (Jones *et al*, 2008; Hemrajani *et al*, 2010; Blasche *et al*, 2013; Li *et al*, 2013; Pearson *et al*, 2013). Intriguingly, some T3SS effectors are widely conserved among enteric bacterial pathogens, irrespective of differences in their infection modalities, e.g., invasive vs. extracellularly attached. For example, OspE homologs are conserved not only in the invasive bacteria *Shigella* and *Salmonella*, but also in the extracellularly attached bacteria EPEC and EHEC, and play similar roles in preventing cell detachment (Miura *et al*, 2006; Kim *et al*, 2009; Morita-Ishihara *et al*, 2013). The OspE effector targets ILK, which resides in the basolateral plasma membrane of host epithelial

**Figure 5.  *Shigella* T3SS effector OspC1-mediated caspase-8 inhibition triggers necroptosis.**

A, B   HT29 cells were infected with *Shigella* WT, *ΔospC1*, *ΔospD3*, *ΔospC1ΔospD3*, or *ΔospC1ΔospD3/ospC1* (*ΔospC1ΔospD3* complemented with *ospC1*) strains and incubated for 8 h. Cell lysates and aliquots of cellular supernatants were subjected to immunoblotting (A) and cytotoxicity assay (B), respectively. *$P < 0.05$; n.s., not significant (one-way ANOVA).

C   HT29 cells were infected with *Shigella* WT, *ΔospD3*, or *ΔospC1*. Cell lysates obtained at the indicated time points were subjected to immunoblotting.

D, E   HT29 cells treated with DMSO or caspase-8 inhibitor were infected with *Shigella* WT, *ΔospC1*, *ΔospD3*, or *ΔospC1ΔospD3* strains and incubated for 8 h. Cell lysates and aliquots of cellular supernatants were subjected to immunoblotting (D) and cytotoxicity assay (E), respectively. *$P < 0.05$; n.s., not significant (unpaired two-tailed Student's *t*-test).

F   HT29 cells treated with the control or caspase-8 siRNAs were infected with the indicated *Shigella* strains and incubated for 8 h. Cell lysates were subjected to immunoblotting. The knockdown efficiency of the indicated siRNAs was assessed by immunoblotting.

Data information: Graphs in (B) and (E) show mean ± SD, and data are pooled from three independent experiments performed in triplicates. Images in (A), (C), (D), and (F) are representative of three independent experiments. Molecular weights in immunoblots are in kDa.
Source data are available online for this figure.

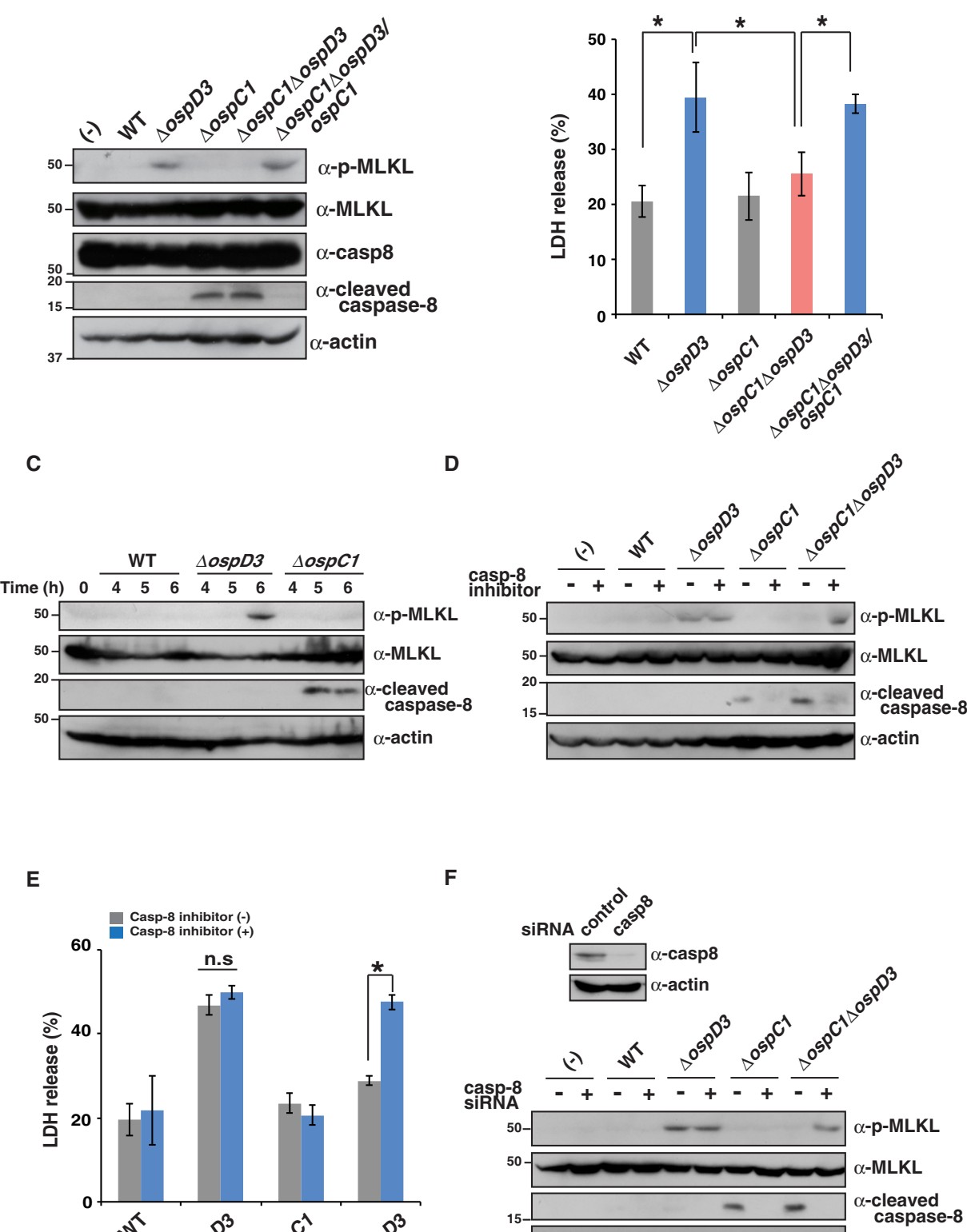

**Figure 5.**

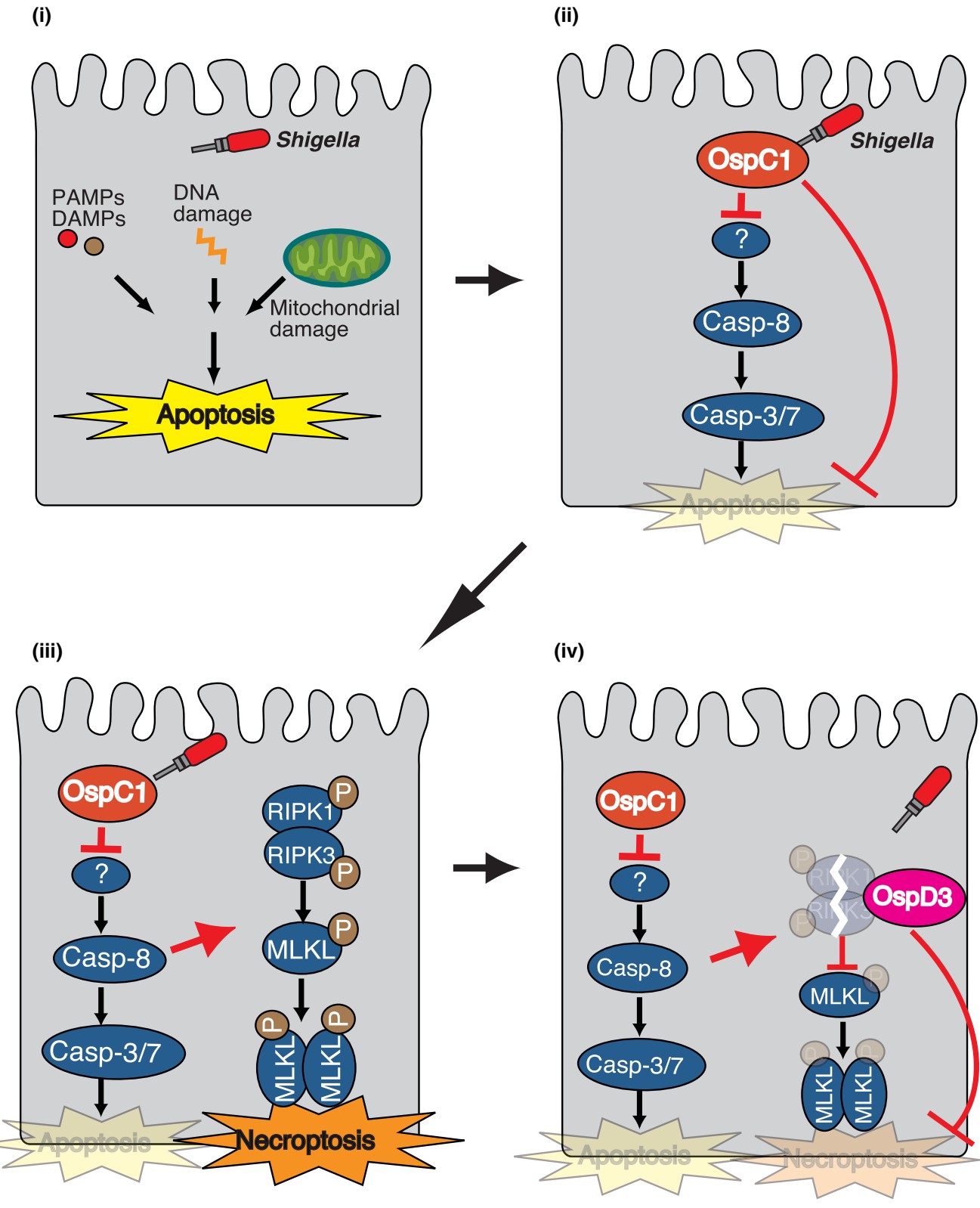

**Figure 6. Model.**

(i) When *Shigella* invades and multiplies within epithelial cells, PAMPS and DAMPs are released. Host cells detect these PAMPs and DAMPs and subsequently trigger apoptosis as host defense to clear bacterial infection. (ii) To counteract this, *Shigella* delivers OspC1 effector and directly or indirectly prevents caspase-8 activation and apoptotic cell death. (iii) On the contrary, host cells detect bacterial disturbance of caspase-8 activation, resulting in induction of necroptosis as a backup host defense. (iv) Again, *Shigella* subsequently delivers OspD3 effector, which targets RIPK1 and RIPK3 for degradation via its protease activity to prevent necroptosis.

cells, and promotes cell adhesion, thereby preventing detachment of infected cells. These studies suggest that bacteria have evolved a universal infectious strategy mediated by widely conserved effectors. Pearson et al, as well as this study, showed that EPEC, EHEC, and *Shigella* OspD3 homologs target RIPK1 and RIPK3 for degradation via their protease activity, leading to inhibition of necroptosis. As with OspE homologs, inhibition of cell death mediated by OspD3 homologs seems to be a conserved infectious strategy among enteric bacterial pathogens with different infection modalities. Therefore, development of novel drugs and vaccines that target OspD3 effectors represents a promising therapeutic strategy that could be used against a wide range of bacterial pathogens.

In summary, this study provides the first evidence that host cells recognize the blockade of caspase-8 apoptosis signaling by the *Shigella* T3SS effector OspC1 and trigger necroptosis as a backup host defense system. Moreover, *Shigella* uses the OspD3 protease to prevent necroptosis by targeting RIPK1 and RIPK3 for degradation. Our findings reveal that bacteria employ previously uncharacterized, temporally regulated tactics to prevent host cell death crosstalk to promote their intracellular survival and multiplication.

# Materials and Methods

### Strain and plasmids

*Shigella flexneri* strain YSH6000 was used as the WT, and S325 (*mxiA*::Tn5) was used as the T3SS-deficient negative control, as described previously (Sasakawa et al, 1986). EPEC E2348/69 was used as the WT strain, and *ΔescF* (T3SS-deficient) was used as the T3SS-deficient negative control. EHEC O157 Sakai was used as the WT strain, and *ΔescF* (T3SS-deficient) was used as the T3SS-deficient negative control. Construction of non-polar mutants of *S. flexneri* YSH6000, EPEC E2348/69, and EHEC O157 Sakai was carried out using the red recombinase-mediated recombination system, as described previously (Datsenko & Wanner, 2000; Ashida et al, 2007). The *ospD3, espL,* and *espL2* coding sequences were amplified by PCR and cloned into pEGFP and pcDNA-Myc$_6$ (6× Myc). *ospD3-FLAG* and *ospD3CS-FLAG* (in which the cysteine residue at position 64 was substituted to serine) were cloned into pWKS130 to yield p-*ospD3-FLAG* and p-*ospD3CS-FLAG,* respectively. The resultant plasmids were introduced into the *ΔospD3* strain. cDNAs for human *RIPK1, RIPK3,* and a series of *RIP* mutants were cloned into pCMV-FLAG and pEGFP vectors. Site-directed mutagenesis of *ospD, espL, espL2,* and *RIPK* was performed using the Quik-Change site-directed mutagenesis kit (Stratagene).

### Materials

Anti-RIP1 (#3493), anti-RIP2 (#4142), anti-RIP3 (#13526), anti-p-RIP1 (#44590), anti-MLKL (#14993), anti-p-MLKL (#91689), anti-caspase-8 (#4790), anti-cleaved caspase-8 (#9496), anti-PARP (#9532), anti-GSDMD (#96458), anti-caspase-8 (#9746), anti-Myc 9B11 (#2276), anti-GFP (#2956) (Cell Signaling Technology), anti-actin (#MAB1501, MILLIPORE), anti-IL18 (PM014, MBL), and anti-FLAG antibodies (F7425, Sigma-Aldrich) were obtained from the indicated suppliers. The following inhibitors, all obtained from Calbiochem, were used at the indicated final concentrations: z-VAD-

FMK, 10 μM; RIP1 kinase inhibitor III, 5 μM; GSK872 (RIPK3 inhibitor), 5 μM; and caspase-8 inhibitor II, 10 μM. Staurosporine was purchased from Sigma.

### Cell culture

Cells were maintained in 5% $CO_2$ at 37°C. HT29 and HCT116 cells were cultured in McCoy's 5A medium (Gibco) supplemented with 10% fetal calf serum. HeLa, HaCaT, and 293T cells were cultured in Dulbecco's modified Eagle medium (Sigma) supplemented with 10% fetal calf serum. Cells were transfected using the FuGENE 6 transfection Reagent (Promega). To construct HeLa cells stably expressing GFP, RIPK3, or RIPK3-4A, cDNAs encoding these genes were subcloned into pMX-puro retroviral expression vectors. Retroviral supernatants were produced in Plat-E cells. HeLa cells expressing the ecotropic viral receptor were transduced with supernatants in the presence of DO-TAP (Roche) and then cloned under puromycin selection.

### Bacterial infection

Cells were infected with various *Shigella* strains expressing afimbrial adhesin (Afa) at a multiplicity of infection (MOI) of 10, as described previously (Ashida et al, 2010). Briefly, infection was initiated by centrifuging the plate at 700 × g for 10 min. After incubation for 30 min at 37°C, the cells were washed three times with PBS and transferred into fresh medium containing gentamicin (100 μg/ml) and kanamycin (60 μg/ml) to kill extracellular bacteria. The time of antibiotic treatment was defined as 0 h after infection. For EPEC and EHEC strains, HT29 cells were infected at an MOI of 50, as described for *Shigella* but with a 1-h incubation. At the indicated times, the cells were washed with PBS and harvested in 2× Laemmli's sample buffer for immunoblotting. For detection of caspase cleavage, infected cells were lysed and mixed with culture medium precipitated with 10% trichloroacetic acid. All samples were separated by SDS–PAGE using either 7.5, 10, or 12% polyacrylamide gels, depending on the molecular weights of the target proteins, followed by immunoblotting.

### Cell death assay

Cells were infected with *Shigella* as described above and then incubated at 37°C. Aliquots of cellular supernatants obtained at the indicated time points were subjected to cytotoxicity assays. Cytotoxicity was analyzed using the CytoTox96 Non-Radioactive Cytotoxicity Assay (Promega). The following formula was used to calculate the amount of LDH release: [($OD_{490}$ sample release − $OD_{490}$ negative control release)/($OD_{490}$ positive control release − $OD_{490}$ negative control release)] × 100, where "$OD_{490}$ negative control release" represents the amount of LDH released into the culture supernatant from uninfected cells, and "OD490 positive control release" represents the amount of LDH released after lysis of the uninfected cells. TUNEL assays were performed with the Dead-End fluorometric TUNEL system (Promega). For PI staining, cells were incubated with PI for 30 min before fixation. Cell nuclei were stained with DAPI. Samples were analyzed on a confocal laser scanning microscope (Carl Zeiss LSM-800), and the percentage of TUNEL-positive or PI-positive cells was calculated from at least 200 cells. For Giemsa

staining, cells were fixed with 4% PFA in PBS; the Giemsa reagent was acquired from Wako. Live images of infected cells were captured on a Nikon ECLIPSE E600 (Nikon). Apoptosis was assayed using the RealTime Glo Annexin V Apoptosis Assay (Promega).

## Caspase activation assay

To measure caspase activity, $5 \times 10^4$ cells were plated in each well of a 96-well plate and stimulated for 8 h with the indicated *Shigella* strains at an MOI of 10. Caspase-1, -3/7, and -8 activities in cell lysates were analyzed using the Caspase-Glo 1, -3/7, and -8 assays (Promega), respectively.

## RNAi

siRNAs against human RIPK1, RIPK3 or caspase-8 with the following sequences, were prepared by Sigma: 5′-GGAUCCGUUAAC-GUUAAUACC-3′ and 5′-UAUUAACGUUAACGGAUCCUG-3′ (RIPK1-#1), 5′-CACCUGUCCGGUUACUACUUG-3′ and 5′-AGUAGUAACCG-GACAGGUGCA-3′ (RIPK1-#2), 5′-GCGGUCAAGAUCGUAAACUCG-3′ and 5′-AGUUUACGAUCUUGACCGCCA-3′ (RIPK3-#1), 5′-GCGACCG-CUCGUUAACAUAUA-3′ and 5′-UAUGUUAACGAGCGGUCGCCC-3′ (RIPK3-#2), 5′-GGAAUGGAACACACUUGGATT-3′ and 5′-UCCAAGU-GUGUUCCAUUCCTG-3′ (caspase-8), and 5′-CGUACGCGGAAUA-CUUCGAUU-3′ and 5′-UCGAAGUAUUCCGCGUACGUU-3′ (luciferase as a control). Cells were transfected using RNAiMax (Invitrogen). siRNA-treated cells were used after 72 h for further analyses.

## Fluorescence microscopy

Cells were washed with PBS and fixed for 20 min with 4% paraformaldehyde in PBS. The fixed cells were immunostained with antibody against cleaved caspase-8 (Cell Signaling Technology), followed by FITC-labeled anti-rabbit IgG (Sigma), rhodamine–phalloidin (Sigma), and DAPI. The samples were analyzed on a confocal laser scanning microscope (Carl Zeiss LSM-800).

## Statistical analysis

Statistical analysis was performed in GraphPad Prism version 6. Differences between two groups were evaluated using unpaired two-tailed Student's *t*-test. One-way ANOVA was used to analyze differences among multiple groups. *P*-values < 0.05 were considered significant.

# Data availability

The authors declare that there are no primary datasets and computer codes associated with this study.

**Expanded View** for this article is available online.

## Acknowledgements
This work was supported by a Grant-in-Aid for Scientific Research (B) (16H05186 to H.A) and AMED under Grant Number 19gm6010009h0003 (H.A). Part of this work was supported by grants from the Naito Foundation (H.A), a research grant from the Astellas Foundation for Research on Metabolic Disorders (H.A), the Uehara Memorial Foundation (H.A), GSK Japan Research Grant 2016 (H.A), the Kawano Masanori Memorial Foundation for Promotion of Pediatrics (H.A), TERUMO FOUNDATION for LIFE SCIENCES and ARTS (H.A), the Senri Life Science Foundation (H.A), the Hamaguchi Foundation for the Advancement of Biochemistry (H.A), the Japan Foundation for Pediatric Research (H.A), and the Takeda Science Foundation (H.A).

## Author contribution
HA designed the project, performed the experiments, interpreted the experimental data, and wrote the manuscript. CS and TS critically revised the manuscript and supervised this study.

## Conflict of interest
The authors declare that they have no conflict of interest.

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
