## [Review Process File · The EMBO Journal]

A unique bacterial tactic to circumvent the cell-death crosstalk induced by blockade of caspase-8

Hiroshi Ashida, Chihiro Sasakawa, and Toshihiko Suzuki
DOI: [10.15252/embj.202010446](https://doi.org/10.15252/embj.202010446)

Review Timeline:

Submission Date:	13th Jan 20
Editorial Decision:	12th Feb 20
Revision Received:	13th Apr 20
Editorial Decision:	14th May 20
Revision Received:	21st May 20
Accepted:	5th Jun 20

Editor: Elisabetta Argenzio

Transaction Report:

Thank you for submitting your manuscript entitled "A unique bacterial tactic to circumvent the cell-death crosstalk induced by blockade of caspase-8" [EMBOJ-2020-104469] to The EMBO Journal. Your study has been sent to three referees for evaluation, whose reviews are enclosed below.

As you can see, the referees find your study interesting and raise only a few points that should be addressed experimentally before they can support the publication of your work in The EMBO Journal. In particular, referee #2 requests you to determine whether separate OspC1- and OspD3-based complexes containing their respective targets exist, and to perform temporal cell death analyses in $\Delta ospC1/2/3ospD3$ cells. Reviewer #3 asks you to include controls for siRNA experiments and test an independent siRNA or to, at least, provide more detail about the control siRNA. In addition, the referees give you suggestions on how to improve the text.

Given the overall interest of your study, I would like to invite you to revise the manuscript in response to the referee reports. I should note that conclusively addressing these and all the other referees' points is essential for publication in The EMBO Journal.

Referee #1:

The article shows that the Shigella OspC1 and OspD3 type III effectors act in concert to prevent the death of infected cells. It is shown that OspC1 inhibits apoptosis mediated by Caspase 8 and that this inhibition is a trigger for the activation of RIPK1-mediated necroptosis, itself counter-acted by OspD3 through RIPK1/3 proteolysis.

RIPK1/3 proteolysis by the EPEC EspL/L2 orthologs had previously been reported as a bacterial strategy to prevent cell death, but this study is first to identify the role for OspC1-mediated Caspase 8 inhibition and the dual OspC1-OspD3 activity in cell death cross-talk. The work has important implications for Shigella pathogenesis and extend the concept of RIPK targeting by pathogens as a general strategy to promote bacterial replication.

This is an elegant and well documented piece of work. The experiments are generally carefully

controlled and the data are convincing. I only have minor comments.

1. Previous works in the literature have shown that OspC1 and OspD3 play a role in the regulation of PMN infiltration and IL-8 secretion during Shigella infection (Farfan et al., 2011; Faherty & al ; 2016). The authors should refer to this literature and integrate / discuss their findings in light of the role of RIPK1 in maintaining inflammatory homeostasis .
2. Fig. 3B shows a smaller RIPK1 fragment suggesting endoproteolysis. Figure 3A suggest disappearance of RIPK1. Please clarify.
3. Can the authors comment on the discrepancy observed for MLKL phosphorylation in cells infected with Shigella Δ ospD3 and EPEC Δ espL/L2 ?
4. Figure 5E : replace annotation of the Δ ospC1/ospD3 lane by Δ ospC1 Δ ospD3. The labeling is misleading as is since it suggests complementation with OspD3.
5. l. 213 : « direct » effect.... Claiming a « direct » effect would implicate the use of purified proteins in an in vitro system. At this point, the authors cannot exclude an « indirect » effect of OspD3, for example, through the activation of an endogenous protease.
6. Careful proofreading of the text would help, a few suggestions below :
7. L. 219 : « Since Pearson [...] syntax problem with the sentence.
8. L. 319 : remove « In fact, ... »
9. L. 221, L. 335 : « Of important. » should read « Of importance »
10. L. 330, add coma : « Although [...] the MLKL phosphorylation and cytotoxicity observed in cells infected with Δ ospD3 was abolished, in cells infected with Δ ospC1 Δ ospD3 ...

Referee #2:

Ashida et al have investigated the mechanisms of how an important bacterial pathogen combats the crosstalk between cell death pathways and ensures the survival of its host cell. They use various mutants of *S. flexneri* and measure cell death using LDH and PI and other assays for caspase-3/7/8 activity, and western blots that reveal phospho-MLKL or GSDMD cleavage as indicators of specific cell death pathways. This is an interesting study that is well executed, includes appropriate assays and controls and data interpretation and siRNA-based experiments alongside data with cell death-inhibitors. The use of natural infection of host cells with bacterial strains in key assays, rather than effector overexpression & treatment with exogenous cytokines, is a plus point of these studies. Given that extremely similar strategy is used by EPEC/EHEC, I am not sure 'unique' is appropriate in the title. Although temporal effects on host cells is shown, whether bacterial effectors are secreted/expressed sequentially is not shown. A few experiments to investigate this will add to the strengths of this study.

The following are suggestions to clarify a few points and, hopefully, will improve the manuscript further.

1. The temporal western blotting and cell death in EV6 is interesting and shows active forms of proteins involved in necroptosis or apoptosis. I feel this panel should be in the main figures with a few additional experiments, perhaps a time point between 4 and 6h and with GSDMD and/or IL-18

blots to further show specificity.

2. Have the authors looked into differential expression and secretion of Shigella effectors (Le Gall et al; <https://doi.org/10.1099/mic.0.27639-0>)? If caspase-8 inhibition occurs early, as the authors suggest, is it because the effector OspC1 is translocated earlier than OspD3? Translocation of OspC1, OspC3 and OspD3 during infection should be investigated.

3. EPEC & EHEC lacking EspL additionally have NleB/NleB1 that modify death domains and prevent the formation of necroptosis-activating signalling complexes (perhaps could be mentioned in the Discussion). As this is not the case with *S.flexneri*, the authors could confirm whether RIPK1/RIPK3 and caspase-8 complexes are already present early and late during infection and whether OspC1 or OspD3 are part of this complex. Are there separate complexes containing OspC1 and its targets and OspD3 and its targets? Immunoprecipitation or similar approaches during infection will indicate whether the effectors cleave/inhibit free RIPK proteins or within complexes that also contain caspase-8 and FADD.

4. The authors confirm previous studies on OspC3 and inhibition of caspase-4-mediated pyroptosis. In addition to the strain lacking ospC1 or ospD3, temporal cell death analyses (like in EV6) should be performed with Δ ospC1/2/3ospD3 to confirm which cell death pathway is activated first e.g. with western blotting for phospho-MLKL, cleaved Casp8 and cleaved GSDMD/IL18. This will clarify which effector needs to act first to prevent host-cell death during infection. Is it possible that caspase-4 is activated even before caspase-8 is (and therefore OspC3 has to act before OspC1)?

Minor points:

1. There are several instances of "Of important" when the authors mean "Of importance". E.g. line 190, 221.

2. Line 219, sentence starting "Since..." needs to be reworded.

3. The authors should generously cite all previous original work on cell death - while Kerr 1972 may be correct, it is not appropriate for the sentences in line 76/77 which mentions GSDMD and caspase-1/11. Please carefully cite work on Shigella, EPEC & EHEC on cell death.

4. I suggest using the human gene naming throughout i.e. say caspase-4 instead of caspase-11. Despite the large number of papers on caspase-11, the updated mouse gene name is also Casp4 (<https://www.ncbi.nlm.nih.gov/gene/12363>).

5. Salmonella infection does cause cell death, so it is not correct to say otherwise (line 84) e.g. PMID: 25121752.

6. The whole study focusses on necrotic cell death, during which gentamicin can gain entry to the cell and bacteria can be released into culture medium containing the antibiotic. I therefore feel that their CFU assays are not appropriate and they do not add value to their findings, and I suggest removing them altogether. If the authors wish to comment on cell death pathways in restricting bacterial replication, they should use live-cell microscopy to confirm reduced bacterial replication before membrane damage (e.g. assays in PMID: 27808091)

Referee #3:

This paper is well written, and the work is logical and easy to follow. Results for ospD3 cleaving RIP1 and RIP3 are as expected, given prior work on *E. coli* by Pearson et al. However, together with the demonstration of ospC1 inhibiting apoptosis this makes a nice package describing a coordinated effort by Shigella to inhibit apoptosis and necroptosis. I have some comments below that the authors should address.

Comments

1. S325, the T3SS mutant is included throughout the paper, but the results for it are never discussed. Please explain to the readers why you included this mutant. My interpretation is that the effectors that induce death are completely dependent on the T3SS. But since I don't know how well it grows in cells, I am not clear on what it is a control for.

It would also help if you can discuss in the introduction about the site of replication and residence of Shigella. I realise it is intracellular, but is it generally membrane bound and using the T3SS to inject effectors into the cytosol from an endosomal compartment, or does it escape significantly into the cytosol? I note that Figure 6 seems to show it in the cytosol rather than within a membrane compartment.

2. All figure legends for quantitative data just say $n=3$, and do not detail at what level replication was done and what the error bars are. Maybe this is OK, if this is always the same, and is detailed in the methods, although if there is room I think it is generally preferable to have these details in the legend. The methods section states "Data are representative of two or three independent experiments. Values are reported as means {plus minus} standard deviation (SD) of data obtained from independent experiments." This is ambiguous regarding whether results presented are actually just a "representative" experiment or whether they summarise all experiments. Some of the error bars look a bit small for a summary of three experiments for things like LDH that are normally quite variable. If these results are just showing replicates from a single experiment, then (i) please show data from all experiments wherever possible; and (ii) doing statistics would not be appropriate on a single experiment. If results are truly summarising 2-3 experiments, please remove the "Data are representative of" and say "Data show results from 2 or 3 independent experiments". If there are only 2 experiments please show range not SD, but I cannot see any figure legends that say " $n=2$ ". So I am confused here, on the replication.

3. I think that the discussion could be improved. Aspects that are not clearly discussed are:

(i) what is the conservation of the proteins you are looking at, amongst bacterial species and how broadly might they be fulfilling this role?

(ii) what are the possible routes of caspase-8 activation during Shigella infection? I realise that this is unknown, but caspase-8 can be activated by death receptors, but also through direct recruitment of caspase-8 to ASC in the inflammasome (Cell Death Diff 2013 20:1149). However I think FAS may be implicated in apoptotic response to EPEC? (Pollock et al. Infection and Immunity 2017 85(4). pii: e01071-16.).

(iii) Regarding the Pollock paper above, are there NleB1 and NleF homologues in Shigella? I think it is useful to discuss their modes of action in E.coli as part of speculation about how OspC1 might function.

4. Regarding gene knockdown with siRNA, I would normally say that due to off target effects at least two siRNA should be used. The best practice is to show cells with no siRNA as well as a control siRNA and at least two gene-specific siRNAs. Since you have looked at several different means of addressing the issue for the RIPK, and everything is consistent, this may not be necessary to insist upon in this case. However, a control siRNA sequence does not seem to be detailed?

5. EspF mutants of E. coli are used in Figure EV3 and not commented on anywhere. Please at least explain why this is included and what it means in the figure legend for this supplementary figure, if it is not important enough to mention in the text.

6. Sometimes the shapes of bands on the western blots do not look like the same blot was always reprobed with different antibodies. If multiple blots had to be run for the probings this should be

explained in the methods.

7. Line 209: "Intriguingly, unlike cells infected with *Shigella* Δ ospD3, cells infected with EPEC Δ espL or EHEC Δ espL2 did not exhibit higher levels of MLKL phosphorylation and cytotoxicity than cells infected with WT (Fig. EV3A and EV3B). Thus, OspD3 homologs can decrease the levels of RIPK1 and RIPK3, but only *Shigella* OspD3 prevents necroptosis." I think this should be rephrased. Pearson et al show that ectopic expression of EspL prevents necroptosis. Clearly if the homologues decrease RIPK1 and 3 they should inhibit necroptosis. Instead of the last sentence I would say "These results suggest that EPEC/EHEC use redundant pathways to inhibit MLKL phosphorylation and necroptosis, and preventing the degradation of RIPK1 and RIPK3 by deletion of EspL proteins is insufficient to restore MLKL phosphorylation".

8. Lines 323-325 needs clearer interpretation. Presumably OspC1 would prevent caspase-8 activation at whatever timepoint it occurs. But the OspD3 mutant shows that the caspase-8 stimulus has only accumulated and commenced acting at 6-8 hours. (This may be a useful point for discussion regarding what might be the stimulus for caspase 8 activation - it takes 6 hours to develop)

Minor points

Lines 344-345. Do you have a reference that shows that apoptosis itself is effective at killing intracellular organisms? This may vary according to the organism, and I would not make this statement unless there is some clear data on this.

Line 314-315 needs rewriting to avoid possible ambiguity. I suggest "Interestingly, unlike for cells infected with Δ ospD3, infection with Δ ospC1 Δ ospD3 did not lead to phosphorylation of MLKL or induction of cell death."

Line 286 should say that the ospC123 mutant showed increased "cleavage" of the caspases and PARP, rather than "activation", which is not directly measured in a western blot.

The concept that when caspase-8 is blocked necroptosis is promoted may be repeated too many times - for example, this is covered in the last paragraph on page 10 and then again in lines 276-77. Just check for repetition of this concept. I think we got it!

Line 103 in the introduction mentions caspase-4, and caspase-11 has been previously mentioned. Please include explanation somewhere the difference between mouse and human caspases in the non-canonical inflammasome pathway.

Line 219: "Since Pearson et al...." is not a grammatical sentence. Not sure what you are saying here.

Western blots need molecular weights indicated

Figure EV1 - staurosporine spelling on graph

Line 540 - typo - Anti-RIP1?

Line 569 "For detection of caspase activation" - please change this to caspase cleavage, and perhaps say culture medium rather than supernatant?

Line 170 should say "To determine whether OspD3 is involved in inhibition of necroptosis..."

Several sentences start with "Of important". This should be "Of importance"

Line 402: I don't think "universal infectious strategy" is appropriate for something that has only been shown in 3 organisms.

Figure 2 legend. Make it clear that you mean "RIPK1 inhibitor" not "RIPK1"

Figure 4 legend: Please change "annexin V activity assay". This is not an activity. You could say annexin V staining for cell membrane phosphatidyl serine exposure. Also change "annexin V activity" To annexin V binding.

Referee #1:

The article shows that the Shigella OspC1 and OspD3 type III effectors act in concert to prevent the death of infected cells. It is shown that OspC1 inhibits apoptosis mediated by Caspase 8 and that this inhibition is a trigger for the activation of RIPK1-mediated necroptosis, itself counteracted by OspD3 through RIPK1/3 proteolysis.

RIPK1/3 proteolysis by the EPEC EspL /L2 orthologs had previously been reported as a bacterial strategy to prevent cell death, but this study is first to identify the role for OspC1-mediated Caspase 8 inhibition and the dual OspC1-OspD3 activity in cell death cross-talk. The work has important implications for Shigella pathogenesis and extend the concept of RIPK targeting by pathogens as a general strategy to promote bacterial replication.

This is an elegant and well documented piece of work. The experiments are generally carefully controlled and the data are convincing. I only have minor comments.

> Thank you very much for your supportive comments on our manuscript. We have taken all of these comments into account and resubmitted a revised version of our paper.

The following are our point-by-point replies to the referee's comments.

1. Previous works in the literature have shown that OspC1 and OspD3 play a role in the regulation of PMN infiltration and IL-8 secretion during Shigella infection (Farfan et al., 2011; Faherty & al ; 2016). The authors should refer to this literature and integrate / discuss their findings in light of the role of RIPK1 in maintaining inflammatory homeostasis .

> According to this comment, we discussed the previous report in the revised text (lines 351–354).

2. Fig. 3B shows a smaller RIPK1 fragment suggesting endoproteolysis. Figure 3A suggest disappearance of RIPK1. Please clarify.

> As pointed out by the referee, cleavage of RIPK1 by OspD3 in *Shigella* infection leads to complete disappearance of the protein (Fig. 3A), whereas cleavage of RIPK1 by transfected OspD3 leads to the appearance of a smaller cleaved protein (Fig. 3B). We attribute the difference in these results to the involvement of the ubiquitin–proteasome system. We believe that stimulants caused by *Shigella* infection, such as PAMPs and DAMPs, trigger activation of a

signaling cascade that targets cleaved RIPK1 for ubiquitination and proteasomal degradation. We plan to address this possibility in future work.

3. Can the authors comment on the discrepancy observed for MLKL phosphorylation in cells infected with Shigella opsD3 and EPEC espL/L2 ?

> As suggested by Referee #3 (please see Comment to Referee #3 No. 7), we rephrased the text (lines 205–208): “These results suggest that EPEC and EHEC use redundant pathways to inhibit MLKL phosphorylation and necroptosis, and that preventing the degradation of RIPK1 and RIPK3 by deletion of EspL proteins is insufficient to restore MLKL phosphorylation.”

Because EPEC and EHEC lacking *espL* also harbor NleB/NleB1 and NleF, which prevent necroptosis-activating signaling, we believe that the discrepancy regarding MLKL phosphorylation in cells infected with *Shigella ΔospD3* and EPEC *ΔespL* reflects a redundant pathway for inhibition of MLKL phosphorylation and necroptosis by EPEC and EHEC. Identification of the differences between *Shigella*, EPEC, and EHEC is an important priority for future work.

4. Figure 5E : replace annotation of the ospC1/ospD3 lane by ospC1ospD3. The labeling is misleading as is since it suggests complementation with OspD3.

> According to this comment, we corrected the annotation of the revised Figure 5E.

5. l. 213 : <direct> effect.... Claiming a <direct> effect would implicate the use of purified proteins in an in vitro system. At this point, the authors cannot exclude an <indirect> effect of OspD3, for example, through the activation of an endogenous protease.

> According to this comment, we removed “direct” in the revised text (line 209).

6. Careful proofreading of the text would help, a few suggestions below :

7. L. 219 : <Since Pearson>... syntax problem with the sentence.

> According to this suggestion, we corrected this sentence in the revised text (line 215-217).

8. L. 319 : *remove <In fact,..>*

> According to this suggestion, we corrected this sentence in the revised text (line 312).

9. L. 221, L. 335 : *<Of important.> should read <Of importance>*

> According to this suggestion, we corrected this sentence in the revised text (line 217).

10. L. 330, *add coma : <Although> the MLKL phosphorylation and cytotoxicity observed in cells infected with ospD3 was abolished, in cells infected with ospC1ospD3 ...*

> According to this suggestion, we corrected this sentence in the revised text (lines 326–329).

Referee #2:

Ashida et al have investigated the mechanisms of how an important bacterial pathogen combats the crosstalk between cell death pathways and ensures the survival of its host cell. They use various mutants of S. flexneri and measure cell death using LDH and PI and other assays for caspase-3/7/8 activity, and western blots that reveal phospho-MLKL or GSDMD cleavage as indicators of specific cell death pathways. This is an interesting study that is well executed, includes appropriate assays and controls and data interpretation and siRNA-based experiments alongside data with cell death-inhibitors. The use of natural infection of host cells with bacterial strains in key assays, rather than effector overexpression & treatment with exogenous cytokines, is a plus point of these studies. Given that extremely similar strategy is used by EPEC/EHEC, I am not sure 'unique' is appropriate in the title. Although temporal effects on host cells is shown, whether bacterial effectors are secreted /expressed sequentially is not shown. A few experiments to investigate this will add to the strengths of this study.

The following are suggestions to clarify a few points and, hopefully, will improve the manuscript further.

> We thank the referee for these supportive comments on our manuscript. We have taken all of these comments into account and resubmitted a revised version of our paper.

We disagree with Referee #2's suggestion that "unique" is not appropriate in the title. As pointed out by another referee, this study is first to identify a role for OspC1-mediated caspase-8 inhibition and dual OspC1–OspD3 activity in cell-death crosstalk. Because this is the first report to reveal a bacterial countermeasure that prevents cell-death crosstalk between apoptosis and necroptosis, we would like to keep "unique" in the title of our revised manuscript.

The following are our point-by-point replies to referee's comments.

1. The temporal western blotting and cell death in EV6 is interesting and shows active forms of proteins involved in necroptosis or apoptosis. I feel this panel should be in the main figures with a few additional experiments, perhaps a time point between 4 and 6h and with GSDMD and/or IL-18 blots to further show specificity.

> According to this suggestion, we performed additional time-course experiments; the results are reported in Fig. 5C and Fig. EV5C. As shown in the revised Fig. 5C, the amount of cleaved caspase-8 in $\Delta ospC1$ -infected cells began to increase after the 5-h time point, whereas the

levels of MLKL phosphorylation in $\Delta ospD3$ -infected cells began to increase after the 6-h time point.

By contrast, the amount of mature IL-18 and cleaved GSDMD in $\Delta ospC3$ -infected cells began to increase after the 2-h time point (revised Fig. EV5C). These temporal analyses of cell death revealed that OspC3-mediated pyroptosis inhibition is followed by OspC1-mediated apoptosis inhibition, and then by OspD3-mediated necroptosis inhibition, during *Shigella* infection. However, as shown in the revised Fig. EV5A, cells infected with the $\Delta ospC3\Delta ospD3$ double mutant still triggered phosphorylation of MLKL, like cells infected with $\Delta ospD3$, suggesting an absence of crosstalk between OspC3-mediated inhibition of pyroptosis and necroptosis.

Taken together, these data further support our hypothesis that *Shigella* T3SS effectors OspC1 and OspD3 blockade host cell-death crosstalk.

2. Have the authors looked into differential expression and secretion of Shigella effectors (Le Gall et al; <https://doi.org/10.1099/mic.0.27639-0>)? If caspase-8 inhibition occurs early, as the authors suggest, is it because the effector OspC1 is translocated earlier than OspD3? Translocation of OspC1, OspC3 and OspD3 during infection should be investigated.

> To address these comments, we first investigated the expression of *ospC1* and *ospD3* mRNA over time during *Shigella* infection.

During growth of *Shigella* in BHI (brain–heart infusion) broth at 37°C, the T3SS is assembled but is not active, and some effectors are stored within the bacterial cells. Once T3SS is activated upon contact with host cells, it promotes secretion of effectors, such as IpaB, and induces transcription of genes encoding a second set of effectors, such as IpaH9.8, under the control of the transcription activator MxiE.

Bacterial mRNAs isolated 2, 4, and 6 h after infection of HT29 cells or cultured in BHI broth at 37°C were quantified by quantitative RT-PCR along with those of *ipaB* and *ipaH9.8* as controls. The levels of mRNA expression were quantified and normalized to that of *rpoA* (a housekeeping gene). As reported previously [Lucchini et al. 2005 (PMID:15618144); Ashida et al. 2007 (PMID:17214743)], the level of *ipaB* mRNA gradually decreased after the bacteria enter host cells, whereas the level of *ipaH9.8* mRNA, which is regulated by MxiE and transcribed after the bacteria enter host cells, gradually increased (Response Fig. 1). On the contrary, the levels of *ospC1* and *ospC3* mRNA increased after bacterial invasion at 2 h and then decreased after 4 h, whereas the level of *ospD3* mRNA continued to increase after bacterial invasion. These

mRNA expression data imply that OspD3 acts at a later stage of infection than OspC1 and OspC3.

To further address the referee's comments, we investigated the translocation of T3SS effectors over time. For this purpose, we designed a peptide antibody that can detect endogenous secretion and translocation of *Shigella* OspC1 and OspD3. We synthesized peptides encompassing residues 33–52 (CVRNAAQQTMPDEKNLKDSAN) of OspC1, 459–470 (CLKYGATSDNKYI) of OspC1, 1–13 (MPSVNLIPSRKIC) of OspD3, or 85–102 (CHHFAFPDEIKNYVSVSEE) of OspD3, coupled the peptides to keyhole limpet hemocyanin with benzidine, and used them to immunize rabbits. From these two pairs of immunizations, however, we did not obtain appropriate antibodies that reacted with endogenous levels of OspC1 and OspD3. Therefore, we could not monitor the translocation of OspC1 or OspD3 over time during *Shigella* infection.

Although we could not detect sequential translocation of OspC1 and OspD3, we believe that the important issue is the timing of caspase-8 activation, rather than T3SS effector translocation. As pointed out by Referee #3 (please see Comment of Referee #3, No. 8), OspC1 would prevent caspase-8 activation whenever it occurs. However, temporal cell-death analysis of $\Delta ospC1$ infection revealed that the caspase-8 stimulus only accumulated and began to act 5 h after *Shigella* infection (see revised Fig. 5C).

Although the stimuli that drive caspase-8 activation during *Shigella* infection remain unclear, future studies will reveal them, along with the mechanisms that lead to cell-death crosstalk between apoptosis and necroptosis during bacterial infection. We intend to address this issue in future work.

Temporal RNA expression analysis of *Shigella* effector gene

Bacterial mRNAs isolated at 2, 4, and 6 h after infection of HT29 cells or cultured in BHI (Brain-heart infusion) broth at 37°C were quantified by quantitative RT-PCR along with those of *ipaB* and *ipaH9.8* as controls. The levels of mRNA expression were quantified and normalized to that of *rpoA* (house keeping gene)

Response Fig. 1 to Referee

3. EPEC & EHEC lacking *EspL* additionally have *NleB/NleB1* that modify death domains and prevent the formation of necroptosis-activating signalling complexes (perhaps could be mentioned in the Discussion). As this is not the case with *S.flexneri*, the authors could confirm whether *RIPK1/RIPK3* and *caspase-8* complexes are already present early and late during infection and whether *OspC1* or *OspD3* are part of this complex. Are there separate complexes containing *OspC1* and its targets and *OspD3* and its targets? Immunoprecipitation or similar approaches during infection will indicate whether the effectors cleave/inhibit free *RIPK* proteins or within complexes that also contain *caspase-8* and *FADD*.

> To address this comment, we performed immunoprecipitation assays in *Shigella*-infected HT29 cells. As mentioned in Comment 2, we could not obtain appropriate antibodies that react with endogenous levels of *OspC1* and *OspD3*. Therefore, we infected HT29 cells with *Shigella* overexpressing C-terminal FLAG tagged-*OspC1* or *OspD3-CS* (in which the cysteine residue at position 64 was replaced with serine). We used the strain overexpressing *OspD3-CS* because *OspD3* targets and degrades *RIP1/RIP3* via its protease activity.

HT29 cells, either non-infected or infected with *Shigella* Δ ospC1/ospC1-FLAG for 6 h, were immunoprecipitated with anti-caspase-8 (Cell Signaling Technology #9746) and Protein G beads (Sigma). Immunoprecipitates were subjected to immunoblotting to detect the apoptosis complex.

We found that caspase-8 precipitated with RIP1, but not OspC1, TRADD, or FADD, in *Shigella*-infected cells (see Response Fig. 2A). Unexpectedly, FADD, an important component of apoptosis complex, did not interact with caspase-8 in cells infected with OspC1-overexpressing *Shigella*. These data prompted us to speculate that OspC1 prevents formation of the apoptosis complex, thereby inhibiting apoptotic signaling by caspase-8.

HT29 cells, either non-infected or infected with *Shigella* Δ ospD3/ospD3-CS-FLAG for 6 h, were immunoprecipitated with anti-RIPK1 (Cell Signaling Technology #3493) and Protein G beads (Sigma). Immunoprecipitates were subjected to immunoblotting to detect the necroptosis complex. As shown in the Response Fig. 2B, RIPK1 precipitated with OspD3-CS and MLKL in *Shigella*-infected cells, indicating that OspD3 cleaves RIPK1 within the necroptosis complex. However, RIPK1 did not precipitate with RIPK3. Because RIPK1 interacts with RIPK3 to trigger necroptosis, we had not anticipated this result, and we plan to investigate this issue further in future work.

However, because we are currently preparing a separate manuscript regarding the molecular mechanism of the OspC1 effector, including the routes of caspase-8 activation during *Shigella* infection and how it affects OspD3 function, we have chosen not to include these immunoprecipitation data in this manuscript.

Interaction between T3SS effector and its targets in *Shigella* infection

- (A) HT29 cells non-infected or infected with *Shigella* Δ ospC1/ospC1-FLAG for 6 h were immunoprecipitated with anti-caspase-8 (Cell Signaling Technology #9746) and Protein G beads (Sigma). Immunoprecipitates were subjected to immunoblotting.
- (B) HT29 cells non-infected or infected with *Shigella* Δ ospD3/ospD3-CS-FLAG for 6 h were immunoprecipitated with anti-RIPK1 (Cell Signaling Technology #3493) and Protein G beads (Sigma). Immunoprecipitates were subjected to immunoblotting.

Response Fig. 2 to Referee

4. The authors confirm previous studies on OspC3 and inhibition of caspase-4-mediated pyroptosis. In addition to the strain lacking ospC1 or ospD3, temporal cell death analyses (like in EV6) should be performed with Δ ospC1/2/3ospD3 to confirm which cell death pathway is activated first e.g. with western blotting for phospho-MLKL, cleaved Casp8 and cleaved GSDMD/IL18. This will clarify which effector needs to act first to prevent host-cell death during infection. Is it possible that caspase-4 is activated even before caspase-8 is (and therefore OspC3 has to act before OspC1)?

> According to this suggestion, we performed a temporal analysis of cell death in cells infected with Δ ospC123 Δ ospD3 (see Response Fig. 3).

The results revealed that the levels of cleaved caspase-8 in Δ ospC123 Δ ospD3-infected cells began to increase after 5 h, whereas the levels of mature IL-18 and cleaved GSDMD began to increase after 2 h post-infection (Response Fig. 3A). Because OspC1-dependent caspase-8

inhibition is essential for triggering necroptosis, infection with $\Delta ospC123\Delta ospD3$ did not lead to phosphorylation of MLKL, in contrast to infection with $\Delta ospD3$ (Response Fig. 3B). However, temporal cell-death analysis of $\Delta ospD3$ infection revealed that the levels of MLKL phosphorylation began to increase after the 6-h time point (revised Fig. 5C). These data indicate that the order of cell-death inhibition by *Shigella* effector is as follows: OspC3-mediated pyroptosis inhibition, OspC1-mediated apoptosis inhibition, and OspD3-mediated necroptosis inhibition.

(A)

(B)

Temporal cell-death analysis during *Shigella* infection

(A) HT29 cells were infected with *Shigella* WT or $\Delta ospC123\Delta ospD3$ strains. Cell lysates obtained at the indicated time points were subjected to immunoblotting.

(B) HT29 cells were infected with *Shigella* WT, S325, $\Delta ospD3$, $\Delta ospC123$, or $\Delta ospC123\Delta ospD3$ strains and incubated for 8 h, and then cell lysates were subjected to immunoblotting.

Response Fig. 3 to Referee

Minor points:

1. There are several instances of "Of important" when the authors mean "Of importance". E.g. line 190, 221.

> We corrected the text accordingly.

2. Line 219, sentence starting "Since..." needs to be reworded.

> We corrected the text accordingly.

3. The authors should generously cite all previous original work on cell death - while Kerr 1972 may be correct, it is not appropriate for the sentences in line 76/77 which mentions GSDMD and caspase-1/11. Please carefully cite work on Shigella, EPEC & EHEC on cell death.

> In response to this comment, we have revised the text (lines 77–78) and the reference.

4. I suggest using the human gene naming throughout i.e. say caspase-4 instead of caspase-11. Despite the large number of papers on caspase-11, the updated mouse gene name is also Casp4 (<https://www.ncbi.nlm.nih.gov/gene/12363>).

> We corrected the text accordingly.

5. Salmonella infection does cause cell death, so it is not correct to say otherwise (line 84) e.g. PMID: 25121752.

> According to this comment, we corrected this sentence in the text (lines 82–84).

6. The whole study focusses on necrotic cell death, during which gentamicin can gain entry to the cell and bacteria can be released into culture medium containing the antibiotic. I therefore feel that their CFU assays are not appropriate and they do not add value to their findings, and I suggest removing them altogether. If the authors wish to comment on cell death pathways in restricting bacterial replication, they should use live-cell microscopy to confirm reduced bacterial replication before membrane damage (e.g. assays in PMID: 27808091)

> We appreciate this supportive comment. Accordingly, we removed the CFU data in the revised Figures 1, 2, and 5.

Referee #3:

This paper is well written, and the work is logical and easy to follow. Results for ospD3 cleaving RIP1 and RIP3 are as expected, given prior work on E. coli by Pearson et al. However, together with the demonstration of ospC1 inhibiting apoptosis this makes a nice package describing a coordinated effort by Shigella to inhibit apoptosis and necroptosis. I have some comments below that the authors should address.

> We thank the reviewer for these supportive comments on our manuscript. We have taken all of these comments into account and resubmitted a revised version of our paper.

The following are our point-by-point replies to the referee's comments.

Comments

1. S325, the T3SS mutant is included throughout the paper, but the results for it are never discussed. Please explain to the readers why you included this mutant. My interpretation is that the effectors that induce death are completely dependent on the T3SS. But since I don't know how well it grows in cells, I am not clear on what it is a control for. It would also help if you can discuss in the introduction about the site of replication and residence of Shigella. I realise it is intracellular, but is it generally membrane bound and using the T3SS to inject effectors into the cytosol from an endosomal compartment, or does it escape significantly into the cytosol? I note that Figure 6 seems to show it in the cytosol rather than within a membrane compartment.

> We used the S325 strain (T3SS-deficient mutant) as a negative control because multiple infectious events during *Shigella* infection, including invasion of and multiplication within epithelial cells, escape from vacuolar membranes, and cell-to-cell spreading, depending on T3SS activity. According to this comment, we added a description of *Shigella* infection modality in the Introduction section of the revised text (lines 93–95) and an explanation of the S325 mutant (line 131).

2. All figure legends for quantitative data just say n=3, and do not detail at what level replication was done and what the error bars are. Maybe this is OK, if this is always the same, and is detailed in the methods, although if there is room I think it is generally preferable to have these details in the legend. The methods section states "Data are representative of two or three independent experiments. Values are reported as means {plus minus} standard deviation (SD) of

data obtained from independent experiments." This is ambiguous regarding whether results presented are actually just a "representative" experiment or whether they summarise all experiments. Some of the error bars look a bit small for a summary of three experiments for things like LDH that are normally quite variable. If these results are just showing replicates from a single experiment, then (i) please show data from all experiments wherever possible; and (ii) doing statistics would not be appropriate on a single experiment. If results are truly summarising 2-3 experiments, please remove the "Data are representative of" and say "Data show results from 2 or 3 independent experiments". If there are only 2 experiments please show range not SD, but I cannot see any figure legends that say "n=2". So I am confused here, on the replication.

> We regret the lack of detailed explanation in the legend. The data are presented as means \pm standard deviation (SD) from triplicate wells. All data are representative of three independent experiments. According to this comment, we corrected the Materials & Method section and the Figure Legends.

3. I think that the discussion could be improved. Aspects that are not clearly discussed are: (i) what is the conservation of the proteins you are looking at, amongst bacterial species and how broadly might they be fulfilling this role?

> To address this suggestion, we added additional discussion to the text (lines 368–374).

(ii) what are the possible routes of caspase-8 activation during Shigella infection? I realise that this is unknown, but caspase-8 can be activated by death receptors, but also through direct recruitment of caspase-8 to ASC in the inflammasome (Cell Death Diff 2013 20:1149). However I think FAS may be implicated in apoptotic response to EPEC? (Pollock et al. Infection and Immunity 2017 85(4). pii: e01071-16.).

> We agree that death receptor signaling is to some extent important for caspase-8 activation during *Shigella* infection. In general, necroptosis can be triggered by stimulation of death receptor family proteins, such as TNFR or FasL, in the absence of caspase-8 activity. To address this issue, we treated cells with TNF inhibitor to confirm the involvement of TNFR signaling.

HT29 cells were infected with the indicated *Shigella* strains in the presence or absence of TNF- α inhibitor (Calbiochem, #654256) and incubated for 8 h. Cell lysates and aliquots of

cellular supernatants were subjected to immunoblotting (see Response Fig. 4). Treatment with TNF- α inhibitor prevented cleavage of caspase-8 in $\Delta ospC1$ -infected cells. These data support the notion that stimulation of TNFR signaling is important for caspase-8 activation during *Shigella* infection.

However, because we are currently preparing a separate manuscript regarding the function of OspC1 effector, we have chosen not to include data regarding the routes of caspase-8 activation during *Shigella* infection in this manuscript.

TNFR signaling is required for caspase-8 activation during *Shigella* infection

HT29 cells were infected with indicated *Shigella* strains in the presence or absence of TNF- α inhibitor, and incubated for 8 h. Cell lysates and aliquots of cellular supernatants were subjected to immunoblotting

Response Fig. 4 to Referee

(iii) Regarding the Pollock paper above, are there NleB1 and NleF homologues in *Shigella*? I think it is useful to discuss their modes of action in *E.coli* as part of speculation about how OspC1 might function.

> As far as we know, *Shigella* does not possess homologs of the NleB and NleF effectors (lines 369–375). We believe that identification of the differences between *Shigella*, EPEC, and EHEC represents an important priority for future work.

4. Regarding gene knockdown with siRNA, I would normally say that due to off target effects at least two siRNA should be used. The best practice is to show cells with no siRNA as well as a control siRNA and at least two gene-specific siRNAs. Since you have looked at several different means of addressing the issue for the RIPK, and everything is consistent, this may not be necessary to insist upon in this case. However, a control siRNA sequence does not seem to be detailed?

> In siRNA knockdown experiments, we used two siRNA and confirmed that they did not have off-target effects. In response to the referee's concern, we provided additional data and information in the revised Figure 2D and 2E, as well as in the Material & Methods section (see RNAi section).

5. EspF mutants of E. coli are used in Figure EV3 and not commented on anywhere. Please at least explain why this is included and what it means in the figure legend for this supplementary figure, if it is not important enough to mention in the text.

> We regret the misspelling: we meant “*escF* (T3SS-deficient mutant of EPEC/EHEC)” rather than “*espF*”. We corrected Fig. EV3 and provided an explanation in the Materials & Methods section.

6. Sometimes the shapes of bands on the western blots do not look like the same blot was always reprobed with different antibodies. If multiple blots had to be run for the probings this should be explained in the methods.

> All samples were separated by SDS-PAGE on 7.5, 10, or 12% polyacrylamide gels, depending on the molecular weights of the target proteins, followed by immunoblotting. To address this comment, we added a description of western blotting (Immunoblotting) in the Materials & Methods section (lines 573–575).

7. Line 209: *"Intriguingly, unlike cells infected with Shigella Δ ospD3, cells infected with EPEC Δ espL or EHEC Δ espL2 did not exhibit higher levels of MLKL phosphorylation and cytotoxicity than cells infected with WT (Fig. EV3A and EV3B). Thus, OspD3 homologs can decrease the levels of RIPK1 and RIPK3, but only Shigella OspD3 prevents necroptosis." I think this should be rephrased. Pearson et al show that ectopic expression of EspL prevents necroptosis. Clearly if the homologues decrease RIPK1 and 3 they should inhibit necroptosis. Instead of the last sentence I would say "These results suggest that EPEC/EHEC use redundant pathways to inhibit MLKL phosphorylation and necroptosis, and preventing the degradation of RIPK1 and RIPK3 by deletion of EspL proteins is insufficient to restore MLKL phosphorylation".*

> We agree, and we have corrected the text (lines 205–208) accordingly.

8. Lines 323-325 needs clearer interpretation. Presumably OspC1 would prevent caspase-8 activation at whatever timepoint it occurs. But the OspD3 mutant shows that the caspase-8 stimulus has only accumulated and commenced acting at 6-8 hours. (This may be a useful point for discussion regarding what might be the stimulus for caspase 8 activation - it takes 6 hours to develop)

> We agree with the referee's comment. As suggested by Referee 2 (see Referee #2 Comment 1), we performed a cell-death time-course experiment. As shown in the revised Fig. 5C, the time course study of caspase-8 activation in Δ ospC1-infected cells revealed that the caspase-8 stimulus began to accumulate after 5 h post-infection. Accordingly, the levels of MLKL phosphorylation in Δ ospD3-infected cells began to increase after the 6-h time point (revised Fig. 5C). These data strongly indicate that OspC1-mediated caspase-8 inhibition triggers necroptosis, which is eventually counteracted by OspD3. To address the referee's comment, we revised the text (lines 315–324).

Minor points

Lines 344-345. Do you have a reference that shows that apoptosis itself is effective at killing intracellular organisms? This may vary according to the organism, and I would not make this statement unless there is some clear data on this.

> According to this suggestion, we corrected this sentence in the text (line 339).

Line 314-315 needs rewriting to avoid possible ambiguity. I suggest "Interestingly, unlike for cells infected with $\Delta ospD3$, infection with $\Delta ospC1\Delta ospD3$ did not lead to phosphorylation of MLKL or induction of cell death."

> According to this suggestion, we corrected this sentence in the text (lines 306–308).

Line 286 should say that the $ospC123$ mutant showed increased "cleavage" of the caspases and PARP, rather than "activation", which is not directly measured in a western blot.

> According to this suggestion, we corrected this sentence in the text (line 279).

The concept that when caspase-8 is blocked necroptosis is promoted may be repeated too many times - for example, this is covered in the last paragraph on page 10 and then again in lines 276-77. Just check for repetition of this concept. I think we got it!

> We revised the text according to this comment.

Line 103 in the introduction mentions caspase-4, and caspase-11 has been previously mentioned. Please include explanation somewhere the difference between mouse and human caspases in the non-canonical inflammasome pathway.

> We followed the suggestions of Referee #2 and Referee #3 (please see Referee #2 Minor comment 4) and revised the text accordingly.

Line 219: "Since Pearson et al...." is not a grammatical sentence. Not sure what you are saying here.

> According to this suggestion, we corrected this sentence in the text (lines 215–217).

Western blots need molecular weights indicated

> According to this suggestion, we added molecular weight indications in the revised Figures.

Figure EV1 - staurosporine spelling on graph

- > According to this suggestion, we revised Fig. EV1.

Line 540 - typo - Anti-RIP1?

- > According to this suggestion, we corrected this sentence in the text (line 548).

Line 569 "For detection of caspase activation" - please change this to caspase cleavage, and perhaps say culture medium rather than supernatant?

- > According to this suggestion, we corrected this sentence in the text (lines 571–573).

Line 170 should say "To determine whether OspD3 is involved in inhibition of necroptosis..."

- > According to this suggestion, we corrected this sentence in the text (line 170).

Several sentences start with "Of important". This should be "Of importance"

- > We revised the text according to this comment.

Line 402: I don't think "universal infectious strategy" is appropriate for something that has only been shown in 3 organisms.

- > According to this suggestion, we corrected this sentence in the text (line 407).

Figure 2 legend. Make it clear that you mean "RIPK1 inhibitor" not "RIPK1"

- > According to this suggestion, we corrected this sentence in the text (line 453–454).

Figure 4 legend: Please change "annexin V activity assay". This is not an activity. You could say annexin V staining for cell membrane phosphatidyl serine exposure. Also change "annexin V activity" To annexin V binding.

> According to this suggestion, we corrected this sentence in the text (lines 495–496).

Thank you for submitting a revised version of your manuscript. It has now been seen by the original referees, whose comments are shown below.

As you will see, referee #1 and #2 find that their criticisms have been sufficiently addressed and support publication of your study in The EMBO Journal. However, referee #3 urges you to perform a proper statistical analysis of the data. In addition to solve this key point from referee #3, there are a few editorial issues concerning text and figures that I need you to address before we can officially accept the manuscript.

Referee #1:

The authors have addressed all my minor suggestions.

Referee #2:

The authors have satisfactorily addressed my comments.

Referee #3:

The paper is improved by a number of changes, and is a nice piece of work. I have a few minor points, mostly introduced when changes were made. But one more major point, which is that the authors have not responded fully to my criticism of their use of statistics on single experiments. The paper overall looks very sound, but the treatment of quantitative data is a deficiency. The authors state that all figures are a single experiment representative of three. They have not explained why they do not average the results of all experiments. I realise that sometimes absolute values vary between experiments, but the differences between samples should remain. At the moment they have done statistics on $n=3$ replicate wells within a single experiment, for every quantitative figure. Replicate wells from one day are not much better than technical replicates for a measurement, and cannot give valid statistics. For any one of these experiments, an error could have been made in estimation of MOI for one strain, and this error would be the same for all three wells. Statistics is only valid in this style of experiment, if done over multiple experiments. The authors should either provide the data of all experiments, or clearly explain why this is not possible. I would recommend showing all experiments, and use of a different symbol for each experiment allows common trends to be discerned. If showing data from a single experiment I do not consider that statistics are valid, and should be removed from all such figures. Furthermore, the EMBO Press author guidelines discourage the use of statistics for when n is small.

Minor points

1. Please provide references to cover lines 92-95
2. Perhaps in line 311 it should say "induction of rapid cell death" since I presume that with deletion of ospC1 and D3 and activation of caspase-8, the cells should still die through apoptosis, and maybe you would detect this if you assayed cell death a bit later?
3. Lines 342-343 "Upon bacterial infection of intestinal epithelial cells, apoptosis facilitates the exclusion of engulfed bacteria and organelles" - not sure what you mean by exclusion. Perhaps you mean "extrusion". I think you could say that "Upon bacterial infection of intestinal epithelial cells, induction of apoptosis is followed by extrusion of the infected cell from the epithelial monolayer". Mentioning organelles in this context is confusing.
4. I presume that LDH release % is generated by comparison with detergent lysis of cells. Please include that detail -i.e. how you establish 100% LDH release.

Referee #3:

The paper is improved by a number of changes, and is a nice piece of work. I have a few minor points, mostly introduced when changes were made. But one more major point, which is that the authors have not responded fully to my criticism of their use of statistics on single experiments. The paper overall looks very sound, but the treatment of quantitative data is a deficiency. The authors state that all figures are a single experiment representative of three. They have not explained why they do not average the results of all experiments. I realise that sometimes absolute values vary between experiments, but the differences between samples should remain. At the moment they have done statistics on n=3 replicate wells within a single experiment, for every quantitative figure. Replicate wells from one day are not much better than technical replicates for a measurement, and cannot give valid statistics. For any one of these experiments, an error could have been made in estimation of MOI for one strain, and this error would be the same for all three wells. Statistics is only valid in this style of experiment, if done over multiple experiments. The authors should either provide the data of all experiments, or clearly explain why this is not possible. I would recommend showing all experiments, and use of a different symbol for each experiment allows common trends to be discerned. If showing data from a single experiment I do not consider that statistics are valid, and should be removed from all such figures. Furthermore, the EMBO Press author guidelines discourage the use of statistics for when n is small.

> We thank the referee for these supportive comments on our manuscript. We have taken all of these comments into account and resubmitted a revised version of our paper.

We regret our misunderstanding of the criticism about statistics. To address the referee's concern, we re-performed statistical analysis of our data in order to average the results of all experiments. We pooled data from three independent experiments (individual experiment was performed in triplicate wells), and calculated mean and standard deviation (SD) as described in a previous report in *EMBO Journal* (Santos et al. 2018 EMBO J [PMID29459437]). We provided new graphs that show mean \pm SD (data are pooled from three independent experiments performed in triplicates). Please see revised Figs. 1A, 1B, 2C–E, 3F, 4A, 4C-F, 5B, 5E, EV1A, EV1C–D, EV2A, EV2E, EV3B, EV5D, and Appendix Fig. S1A. Statistical analysis was performed in GraphPad Prism version 6. Differences between two groups were evaluated using unpaired two-tailed Student's *t*-test. One-way ANOVA was used to analyze differences among multiple

groups. None of these changes in the statistical analyses affected our results or conclusion. We hope you agree that the revised version of the manuscript is much improved.

Minor points

1. Please provide references to cover lines 92-95

> According to this comment, we provided an additional reference in line 96.

2. Perhaps in line 311 it should say "induction of rapid cell death" since I presume that with deletion of ospC1 and D3 and activation of caspase-8, the cells should still die through apoptosis, and maybe you would detect this if you assayed cell death a bit later?

> According to this suggestion, we corrected this sentence in the text (lines 312).

3. Lines 342-343 "Upon bacterial infection of intestinal epithelial cells, apoptosis facilitates the exclusion of engulfed bacteria and organelles" - not sure what you mean by exclusion. Perhaps you mean "extrusion". I think you could say that "Upon bacterial infection of intestinal epithelial cells, induction of apoptosis is followed by extrusion of the infected cell from the epithelial monolayer". Mentioning organelles in this context is confusing.

> We agree, and we have corrected the text (line 343–345) accordingly.

4. I presume that LDH release % is generated by comparison with detergent lysis of cells. Please include that detail -i.e. how you establish 100% LDH release.

> To address this comment, we added detailed information about the cytotoxicity assay in the Materials & Methods section (lines 481–486).

I am pleased to inform you that your manuscript has been accepted for publication in The EMBO Journal.

Corresponding Author Name: Hiroshi Ashida, Toshihiko Suzuki

Manuscript Number: EMBOJ-2020-104469R